# The Curse of Multi-Modalities: Evaluating Hallucinations of Large Multimodal Models across Language, Visual, and Audio

**Sicong Leng** [1,2*]  **Yun Xing** [2*]  **Zesen Cheng** [1,4*]  **Yang Zhou** [3]  **Hang Zhang** [1,4]
**Xin Li** [1,4]  **Deli Zhao** [1,4]  **Shijian Lu** [2,†]  **Chunyan Miao** [2]  **Lidong Bing** [1,4]

[1] DAMO Academy, Alibaba Group    [2] Nanyang Technological University
[3] IHPC A*STAR, Singapore    [4] Hupan Lab, Hangzhou, China

Project Page: `https://cmm-damovl.site/`
Data: `https://huggingface.co/datasets/DAMO-NLP-SG/CMM`
Code: `https://github.com/DAMO-NLP-SG/CMM`

## Abstract

Recent advancements in large multi-modal models (LMMs) have significantly enhanced performance across diverse tasks, with ongoing efforts to further integrate additional modalities such as video and audio. However, most existing LMMs remain vulnerable to hallucinations, the discrepancy between the factual multi-modal input and the generated textual output, which has limited their applicability in various real-world scenarios. This paper presents the first systematic investigation of hallucinations in LMMs involving the three most common modalities: language, visual, and audio. Our study reveals two key contributors to hallucinations: over-reliance on unimodal priors and spurious inter-modality correlations. To address these challenges, we introduce the benchmark The **C**urse of **M**ulti-**M**odalities (**CMM**), which comprehensively evaluates hallucinations in LMMs, providing a detailed analysis of their underlying issues. Our findings highlight key vulnerabilities, including imbalances in modality integration and biases from training data, underscoring the need for balanced cross-modal learning and enhanced hallucination mitigation strategies. Based on our observations, we suggest potential research directions that could enhance the reliability of LMMs.

## 1 Introduction

Large Multi-modal Models (LMMs) have rapidly advanced, driving significant improvements across a wide range of tasks by effectively integrating and processing diverse data modalities. These models [28, 70, 21, 64, 55, 2, 52, 41], leveraging multi-modal inputs such as image and text, have achieved notable performance gains, particularly in generating contextually accurate textual outputs. As the field evolves, there is a growing trend toward incorporating additional modalities, such as audio and video [62, 8, 58, 69, 11, 29, 25, 51, 18, 13], to enhance LMMs' ability to understand and interact with complex real-world environments.

However, despite these advancements, LMMs are prone to hallucination, a critical issue where generated outputs do not accurately reflect the multi-modal inputs [35, 54, 59, 40]. This issue severely

---

[*]Equal contribution.
[†]Corresponding author.

39th Conference on Neural Information Processing Systems (NeurIPS 2025) Track on Datasets and Benchmarks.

undermines the reliability and applicability of LMMs in real-world scenarios, particularly in tasks requiring precise and factual content generation. Hallucination, particularly object hallucination, has been a key focus in LMMs that handle image and text inputs. Object hallucination occurs when LMMs generate semantically coherent but factually unaligned contents with the actual objects present in the input images. Various benchmarks [31, 49, 38, 54] and mitigation techniques have been proposed to address this issue by refining training processes [35], implementing post-hoc correction [27, 71], etc. However, accommodating additional modalities like audio and video exacerbates alignment and fusion difficulties [26, 14, 53, 33], which lead to increased hallucinations.

This study systematically examines how LMMs produce hallucinations while integrating language, visual, and audio inputs, revealing the prevalence and causes of hallucinations under such multi-modal scenarios. Two key contributors are identified: (1) *over-reliance on unimodal priors*: Models over-rely on data from a single modality, neglecting others. This results in outputs that do not accurately reflect the full range of input data, as models default to familiar patterns within one modality despite contradictory signals from others. (2) *spurious inter-modality correlations*: Models learn erroneous cross-modal associations based on patterns that appear statistically significant but lack meaningful or causal connections, leading to plausible but counterfactual outputs. We introduce The **C**urse of **M**ulti-**M**odalities (**CMM**), a comprehensive benchmark for assessing hallucinations in LMMs, covering a wide range of scenarios across visual, audio, and their joint contexts. CMM converts hallucination evaluation into a binary classification task through object-level and event-level probing. It comprises $1,200$ video/audio/video-audio samples across various multi-modal contexts, ensuring balanced evaluation with $2,400$ probing questions evenly split between queries for existent and non-existent objects/events. LMMs are prompted with straightforward yes-or-no questions regarding the presence of objects or events in the input modalities.

CMM is the first benchmark to systematically investigate LMMs' hallucinations in such comprehensive multi-modal settings. Unlike prior benchmarks that broadly assess hallucination performance, CMM segments hallucinations into nuanced subcategories under two key contributors: *spurious inter-modality correlations* (e.g., visual-language, audio-language, visual-audio-language) and *unimodal over-reliance* (e.g., language domiance, visual dominance, audio dominance), enabling precise diagnosis of LMM vulnerabilities and shedding light on possible improvements. By introducing diagnostic metrics including perception accuracy (PA) and hallucination resistance (HR), CMM offers a comprehensive framework for gauging both perception capabilities and hallucination severity in LMMs. In summary, the contributions of this work are threefold:

- We conduct the first systematic investigation of hallucinations in LMMs across language, visual, and audio modalities, identifying their key contributors including unimodal prior over-reliance and spurious inter-modality correlations.
- We introduce a novel and comprehensive benchmark, The **C**urse of **M**ulti-**M**odalities (**CMM**), which evaluates hallucinations using object-level and event-level probing within a binary classification framework. CMM defines hallucinations with nuanced subcategories and granularities, enabling comprehensive diagnosis of LMM vulnerabilities across various modalities.
- We evaluate a diverse set of state-of-the-art LMMs across visual, audio, and joint contexts, revealing critical insights in model limitations and fundamental challenges in multi-modal learning. Our thorough analysis and discussion pinpoint future directions for mitigating hallucinations and enhancing LMM reliability, providing a viable roadmap for improvements.

## 2   Analyzing Hallucinations across Language, Visual, and Audio

This section systematically investigates the underlying causes of hallucinations in Large Multi-modal Models (LMMs). It includes qualitative demonstrations and comprehensive statistical analysis from two key perspectives: *over-reliance on unimodal priors* and *spurious inter-modality correlations*. Our analysis provides empirical evidence and quantifies the extent to which these factors influence LMMs' reliability.

**Notations.**  Consider an LMM parametrized by $\theta$ that processes inputs from three modalities: language $x$, visual $v$, and audio $a$. The model generates textual output $y$ autoregressively, where each token $y_t$ is conditioned on all three modalities and the previously generated tokens $y_{<t}$:

$$y_t \sim p_\theta(y_t \mid v, a, x, y_{<t}),$$

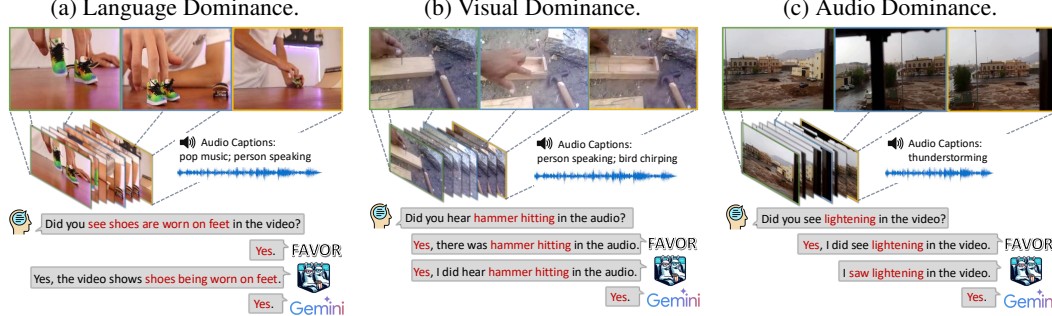

Figure 1: Demonstrations of over-reliance on unimodal priors.

where $y_t$ represents the token at time step $t$, and $y_{<t}$ denotes the sequence of tokens generated up to time step $t - 1$.

## 2.1 Over-reliance on Unimodal Priors

Over-reliance on unimodal priors is a key factor contributing to hallucinations in LMMs. This issue arises when the model over-relies on the knowledge learned from one modality during training, rather than integrating knowledge of all available modalities. In such cases, the model defaults to strong unimodal priors learned during training, leading to outputs that follow familiar unimodal patterns even when those patterns are not supported by the multi-modal input. Following the general issue of over-reliance on unimodal priors, we categorize this into three distinct types: Language Dominance, Visual Dominance, and Audio Dominance. Each form of dominance presents unique challenges for LMM performance and contributes to hallucinations in different ways.

**Language Dominance**. Also known as language biases [42, 27, 19, 56], language dominance arises when a model excessively depends on pre-trained large language models (LLMs), generating responses that adhere to linguistic patterns or prior knowledge from large language corpora, even when visual or audio inputs provide contradictory information. This issue is particularly prevalent in LMMs that integrate LLMs as their decoder base. These LLMs [12, 23, 63], due to their proficiency in capturing linguistic structures and semantic relationships, often dominate the decision-making process, overshadowing contributions from visual or audio modalities. As illustrated in Fig. 1a, a video depicts finger skateboarding with shoes on fingers. When asked by the language-biased question "Did you see shoes worn on feet?"—reflecting commonsense event that follows linguistic priors—LMMs respond "yes", contradicting the actual content and inducing hallucination. This demonstrates LMMs' tendency to rely on language priors over factual multi-modal inputs.

**Visual Dominance**. This occurs when a model over-relies on visual information, with less considering linguistic and auditory cues. In such cases, the model outputs are dominated by visual context, neglecting important information from other input modalities. As illustrated in Fig. 1b, a video depicts a person planning a woodworking project with a hammer in sight, while the audio track contains only events of person speaking and bird chirping. Despite this, advanced LMMs may over-rely on the visual presence of the "hammer" and incorrectly infer a "hammer hitting" sound, ignoring the actual audio content where no such sound is present.

**Audio Dominance**. This phenomenon arises when a model is excessively relying on auditory input, disregarding visual or linguistic inputs. As illustrated in Fig. 1c, a video captures a person recording a village view through a window, showing dark clouds. The audio track contains evident thunderstorm sounds, but no lightning is visible. Despite this, LMMs may over-rely on the audio cues, hallucinating that lightning is visible in the scene, thereby disregarding the actual visual content.

**Case Study**. To further explore our observations on unimodal over-reliance, we performed case studies on each example in Fig. 2, hypothesizing that gradually altering information from a dominant modality would significantly affect the model's responses if hallucinations are primarily due to over-reliance on that modality[3].

---

[3]To further validate unimodal over-reliance, we conduct similar experiments using open-source models across more samples in Appendix A.1.

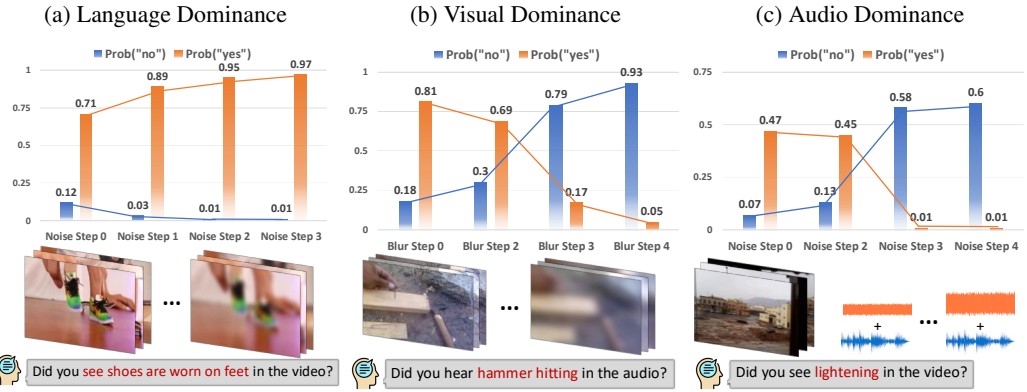

Figure 2: Validation experiments on over-reliance on unimodal priors.

In the visual dominance scenario, we progressively blur the video to reduce visual content and tracked the probabilities of the LMM responding with a hallucinatory "yes" ($p_\theta$("yes" | $v', a, x$)) or a correct "no" ($p_\theta$("no" | $v', a, x$)) across different blur levels. As shown in Fig. 2b, increasing the blur led to a significant decline in hallucinatory "yes" responses and a rise in correct "no" responses. This indicates that reducing visual information compels the model to rely more on auditory cues, thereby decreasing visual-induced hallucinations. In the audio dominance case (Fig. 2c), we add noise to the audio track to degrade its quality. As noise levels increased, the probability of hallucinatory "yes" responses decreased, while correct "no" responses became more frequent ($p_\theta$("yes"/"no" | $v, a', x$)). This demonstrates that diminishing auditory information shifts the model's reliance to visual cues, mitigating hallucinations caused by over-reliance on auditory inputs. For the language dominance scenario, we blur the video containing critical visual information needed to accurately answer an adversarial question. As the visual content was increasingly obscured, the model's reliance on language priors intensifies, leading to more hallucinatory "yes" responses and fewer correct "no" responses (Fig. 2a). This suggests that in the absence of visual details, the model defaults to language-based patterns, exacerbating hallucinations.

In summary, these case studies confirm that unimodal over-reliance significantly contributes to hallucinations in LMMs. Reducing information from the dominant modality forces the model to integrate cues from other modalities more effectively, thereby decreasing the likelihood of hallucinations. This validates the challenges posed by uni-modality over-reliance in multi-modal integration.

## 2.2 Spurious Inter-modality Correlations

Spurious inter-modality correlations are a major contributor to hallucinations in LMMs, especially when integrating multiple modalities. Learned during pretraining on large-scale multi-modal datasets (e.g., image-caption, video-caption, and audio-caption data [34, 24, 6, 43, 57, 48]), these correlations involve misleading associations between modalities that appear statistically significant but lack meaningful or causal connections. Two common sources of spurious correlations are: (1) *Global occurrence frequency*. The high overall occurrence of specific objects or events in the dataset leads LMMs to hallucinate these elements even when they are absent in the input. (2) *Co-occurrence frequency*. Frequent co-occurrence of objects or events during training causes the model to incorrectly predict the presence of one of them when only the other is present. While spurious object-level correlations between language and visual inputs have been extensively studied [42, 31, 71], integrating additional modalities like audio introduces new complexities, resulting in increasingly intricate spurious correlations. We categorize them into three subtypes:

**Visual-Language**. The model hallucinates visual objects or events based on pre-training patterns. For instance, if "phone" frequently co-occurs with "human" in captions, the model may hallucinate a phone upon recognizing a human, even when no phone is present.

**Audio-Language**. The model links absent sound events to textual descriptions due to over-represented patterns in pre-training data. For example, if "dog barking" frequently appears during pre-training, the model may hallucinate this audio event even when the dog in the current input simply whimpers.

| (a) Visual-Language | (b) Audio-Language | (c) Visual-Audio |
|:---:|:---:|:---:|
| 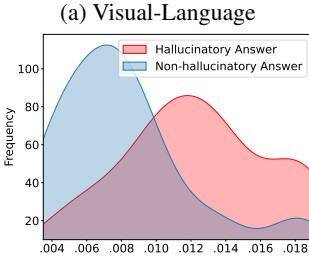 | 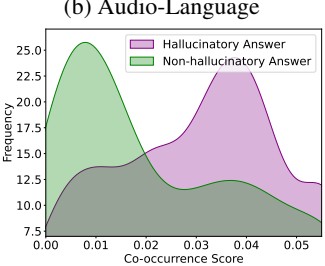 | 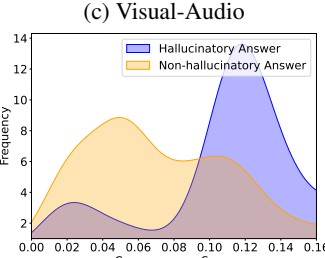 |

Figure 3: Validation experiments on spurious inter-modality correlations caused by co-occurrences.

**Visual-Audio-Language**. Spurious correlations arise from frequent co-occurrence of visual objects and audio events in video-audio joint training. For example, if "bird chirping" in audio descriptions is often paired with "tree" in visual annotations, the model may hallucinate to see trees when only hearing birds, or vice versa.

**Quantitative Analysis**. To validate spurious inter-modality correlations, we curate 200 samples for each subtype, paired with probing questions that target non-existent objects or events based on learned co-occurrence patterns. For VL, video-only samples are paired with questions about non-existent objects that frequently co-occur. In AL, all queries are event-level, targeting absent audio events while a co-occurring action-object pair is present (e.g., querying "dog barking" when the dog only whimpers). For VAL, video-audio pairs are probed for non-existent visual objects or audio events based on frequently co-occurring pairs. We adopt the Co-occurrence Score (CoScore) from previous work [7, 71] to quantify co-occurrence frequency:

$$\text{CoScore}_s = \sum \frac{|S(o_{s,i}) \cap S(o_{s,j})|}{|S(o_{s,i})| + |S(o_{s,j})|},$$

where $S(o_{s,i})$ denotes the set of captions mentioning the $i$-th object or event within a sample $s$. Three open-source LMMs (FAVOR [46], GroundingGPT [32], VideoLLaMA2-7B [11]) are evaluated, with aggregated results shown in Fig. 3, plotting CoScore against the frequency of hallucinatory and non-hallucinatory answers. A consistent trend emerges: hallucinatory responses are associated with higher CoScores, indicating that higher co-occurrence frequencies increase the likelihood of hallucinations. This confirms the impact of spurious inter-modality correlations learned during pretraining[4].

## 3  CMM Benchmark

Table 1: Overview of CMM subcategories.

| | Spurious Inter-modality Correlations | | | Unimodality Over-reliance | | |
|:---|:---:|:---:|:---:|:---:|:---:|:---:|
| **Subcategories** | Visual-Language | Audio-Language | Visual-Audio-Language | Visual Dominance | Audio Dominance | Language Dominance |
| **Input Modality** | Visual | Audio | Visual+Audio | Visual+Audio | Visual+Audio | Visual |
| **Granularity** | object-, event-level | event-level | object-, event-level | event-level | object-level | object-, event-level |
| **# Samples** | 400 | 400 | 400 | 400 | 400 | 400 |

Inspired by the findings in previous section, we introduce The **C**urse of **M**ulti-**M**odality (**CMM**) benchmark, designed to systematically evaluate hallucinations in LMMs from two key contributors: over-reliance on unimodal priors and spurious inter-modality correlations. As shown in Tab. 1, each type is further divided into specific sub-categories, enabling fine-grained assessment of how these factors influence LMMs' performance[5].

### 3.1  Data Composition and Evaluation Setup

For each subcategory, we manually collect 200 samples (video-only, audio-only, or video-audio pairs) to evaluate LMMs' handling of multi-modal inputs. Each sample includes two modality-specific probing questions: one targeting a non-existent object or event (ground-truth answer "no") and one targeting an existent object or event (ground-truth answer "yes"): This results in a total of

---

[4]Further experimental details are provided in the Appendix A.

[5]More benchmark details such as video/audio lengths and object/event distributions are provided in Appendix C.1.

$1,200$ samples and $2,400$ probing questions. We benchmark LMMs using two core metrics, namely, Perception Accuracy (PA) and Hallucination Resistance (HR):

$$PA = \frac{\#\text{correct ``yes''}}{\#\text{ground-truth ``yes''}}, \quad HR = \frac{\#\text{correct ``no''}}{\#\text{ground-truth ``no''}}$$

PA measures the model's ability to accurately perceive present objects or events, while HR assesses its resistance to hallucinations by correctly identifying the absence of non-existent objects or events.

## 3.2 Data Construction

### 3.2.1 Over Reliance on Unimodal Priors

To assess over-reliance on a single modality (visual, audio, or language), we construct targeted probing queries that test the model's dependence on one modality while ignoring complementary signals.

**Visual Dominance**. This subcategory tests whether LMMs hallucinate audio events based on visual input, where we construct queries asking about the existence of specific audio event. While queries with a "yes" answer are manually annotated, non-existent events are sourced from video-audio pairs in the AudioCaps dataset [24], where visual objects that do not correspond to any audio content are identified. These samples are manually verified to ensure accurate annotation, setting the ground truth answer as "no."

**Audio Dominance**. We probe LMMs' tendency to infer incorrect visual content from audio cues. Queries ask about the presence of visual objects, with "yes" queries annotated manually. For "no" queries, we filter video-audio pairs from AudioCaps where audio-indicated objects have no visual representation, confirmed through manual review.

**Language Dominance**. To explore how language priors contribute to hallucinations, we define sets of common-sense events (e.g., "fish swim in water") and object attributes (e.g., "yellow banana") to reflect typical linguistic biases. Videos are manually sourced from YouTube to depict anti-common-sense scenarios (e.g., "fish fly in the air," "black banana"). For existence-probing queries, we ask about the video's anti-common-sense object/event, annotating the ground truth as "yes." Conversely, for non-existence probing queries, we test for the common-sense version of the object/event, setting the ground truth as "no."

### 3.2.2 Spurious Inter-modality Correlations

We evaluate hallucinations arising from *spurious inter-modality correlations*, constructing object-level and event-level queries across visual, audio, and textual associations[6].

**Visual-Language**. Hallucinations are assessed based on associations between visual content and textual descriptions. Object-level queries are derived from (i) global appearance frequencies and (ii) co-occurrence frequencies within the video-caption dataset. Event-level queries, however, are constructed based on (i) global appearance patterns and (ii) [subject]-predicate-object] co-occurrence patterns. All samples are curated from WebVid-10M [6].

**Audio-Language**. Hallucinations derived from associations between audio and text are probed through event-level queries, given the temporal nature of audio. Queries are formed from (i) global appearance frequencies and (ii) subject-oriented co-occurrence patterns, based on data from the Auto-ACD [48].

**Visual-Audio-Language**. This subcategory explores hallucinations across visual and audio modalities. Queries probe non-existent audio events based on existent co-occurred visual objects and vice versa, with data sourced from AudioCaps [24], focusing on co-occurrence frequencies between visual objects and audio events.

---

[6]For more details on the construction process and data statistics, please refer to Appendix C.2.

Table 2: Benchmarking proprietary and open-source Audio-Visual LMMs on CMM.

| Model | Spurious Inter-modality Correlation | | | | | | Uni-modality Over-reliance | | | | | | Overall | |
| | VL | | AL | | VAL | | V | | A | | L | | | |
| | pa | hr | pa | hr | pa | hr | pa | hr | pa | hr | pa | hr | pa↑ | hr↑ |
| Proprietary Models | | | | | | | | | | | | | | |
| Gemini-1.5-Pro | 91.0 | 90.5 | 94.0 | 14.5 | 86.0 | 67.0 | 82.5 | 34.0 | 90.5 | 82.0 | 78.5 | 61.5 | 87.1 | 58.3 |
| Gemini-2.0-Flash | 95.0 | 83.5 | 98.5 | 47.0 | 97.5 | 68.0 | 96.5 | 36.0 | 93.0 | 71.0 | 94.0 | 62.5 | **95.8** | 61.3 |
| Gemini-1.5-Flash | 93.5 | 90.0 | 88.5 | 39.5 | 88.5 | 70.5 | 79.0 | 36.5 | 90.5 | 86.5 | 90.5 | 62.0 | 88.4 | 64.2 |
| Reka-Core | 87.0 | 94.5 | 25.0 | 76.0 | 76.7 | 85.1 | 35.6 | 69.4 | 80.8 | 82.7 | 75.0 | 76.0 | 63.7 | **80.9** |
| Open-Source Models | | | | | | | | | | | | | | |
| GroundingGPT | 95.5 | 36.5 | 100. | 0.0 | 97.5 | 18.0 | 99.5 | 1.0 | 98.5 | 23.5 | 88.5 | 7.0 | **96.6** | 14.3 |
| PandaGPT | 96.5 | 27.0 | 90.5 | 11.0 | 84.5 | 17.5 | 89.0 | 13.5 | 95.0 | 17.5 | 87.0 | 18.5 | 90.5 | 17.5 |
| OneLLM | 97.0 | 67.0 | 97.5 | 17.0 | 97.5 | 34.5 | 95.5 | 11.0 | 99.0 | 8.0 | 75.0 | 32.0 | 93.7 | 28.3 |
| FAVOR | 91.0 | 55.0 | 94.5 | 45.0 | 94.5 | 69.0 | 89.0 | 21.5 | 92.0 | 43.5 | 92.0 | 18.5 | 92.2 | 42.1 |
| VideoSalmonn | 60.0 | 71.5 | 70.0 | 89.0 | 67.0 | 80.0 | 59.5 | 90.0 | 61.0 | 51.5 | 59.0 | 30.5 | 62.8 | 68.8 |
| VideoLLaMA 2 | 75.0 | 86.0 | 77.5 | 94.0 | 78.0 | 98.0 | 62.0 | 75.5 | 80.0 | 90.0 | 57.5 | 43.0 | 71.7 | 81.1 |
| Qwen2.5-Omni | 88.5 | 97.0 | 92.0 | 83.5 | 91.0 | 97.5 | 89.5 | 74.5 | 82.5 | 80.0 | 68.5 | 85.0 | 85.3 | **86.3** |

# 4 Experiments and Discussions

## 4.1 Implementation Details

**Models**. We evaluate a diverse set of LMMs on our benchmark, categorized into three groups based on their modality capabilities: models capable of processing both visual and audio inputs, visual-only models, and audio-only models.

- **Audio-Visual LMMs**. We test a wide range of audio-visual LMMs that are capable of handling video-audio input pairs simultaneously, including proprietary models from [41, 52] and existing open-source models [32, 44, 47, 20, 11, 45, 61] [7].
- **Visual-Only LMMs**. These include models that only take video as inputs without paired audios, including [29, 8, 62, 21, 70, 28, 5]. We evaluate two subsets from CMM for these models, specifically, Visual-Language Spurious Correlation and Visual Dominance.
- **Audio-Only LMMs**. These include models that only takes audio as inputs, including [25, 51, 18, 13]. We evaluate Audio-Language Spurious Correlation and Audio Dominance.

**Evaluation Protocol**. All models are evaluated using a sampling decoding strategy with a fixed temperature of 0.2 for consistency. We assess models based on Perception Accuracy (PA) and Hallucination Resistance (HR) metrics (see Sec. 3.1). All models are post-prompted to *answer with yes or no*, where PA and HR are computed based on whether their responses include *yes* or *no*, following [31].

## 4.2 Main Results

### 4.2.1 Analyzing Audio-Visual LMMs

The results of LMMs that can process both visual and audio inputs are presented in Tab. 2.

**Hallucination Vulnerability from Spurious Inter-Modality Correlations**. Visual-Audio LMMs generally achieve PA scores over 80, demonstrating effective multi-modal perception. Extensive efforts to mitigate Visual-Language (VL) spurious correlations have significantly reduced hallucinations, as proprietary models like Reka-core and Gemini-1.5 reach HR scores around 90. In contrast, open-source models like FAVOR and GroundingGPT continue to struggle with VL correlations. However, the introduction of audio intensifies hallucinations across all models. Even Gemini-1.5-Pro only attains a 14.5 HR score for Audio-Language (AL) correlations, highlighting the difficulty in handling these correlations. Moreover, AL correlations cause more severe hallucinations than Visual-Audio-Language (VAL) correlations, likely due to the limited availability of visual-audio-language datasets compared to audio-language data. This imbalance may lead LMMs to form stronger spurious correlations between audio and language, leading to more frequent hallucinations when processing audio-only content.

---

[7]Omni-LMMs incapable of handling tri-modalities simultaneously are not included, such as [1, 17, 30].

Table 3: Results for Visual-language and Audio-Language.

| Model | VL Cor | | Lan Dom | |
|---|---|---|---|---|
| | pa | hr | pa | hr |
| CogVLM2-Video | 99.5 | 44.0 | 98.0 | 5.0 |
| VideoChat2 | 97.0 | 66.0 | 88.0 | 34.5 |
| InternLM-XC2.5 | 99.0 | 73.0 | 94.5 | 46.5 |
| PLLaVA | 89.5 | 93.0 | 75.0 | 52.0 |
| ShareGPT4Video | 87.5 | 85.5 | 79.5 | 58.0 |
| LLaVA-OV | 94.0 | 88.0 | 87.5 | 69.5 |
| Qwen2.5VL | 89.0 | 97.0 | 66.5 | 87.0 |

(a) Visual-Language LMMs results.

| Model | AL Cor | |
|---|---|---|
| | pa | hr |
| Qwen2-Audio | 98.5 | 34.5 |
| Audio-Flamingo | 89.5 | 39.0 |
| GAMA-IT | 94.5 | 52.0 |
| SALMONN | 93.0 | 59.0 |

(b) Audio-Language LMMs results.

**Hallucination Vulnerability from Uni-modality Over-reliance.** Models show solid perception capabilities across Uni-modality Over-reliance subcategories, with high PA scores. However, a notable gap emerges when comparing PA and HR scores, highlighting hallucination challenges due to unimodal dependence. Visual Dominance, in particular, proves to be more problematic than Audio Dominance for most models. For instance, Gemini-1.5-Flash achieves an HR of $86.5$ in Audio Dominance but only $36.5$ in Visual Dominance, suggesting that over-reliance on visual input presents a more significant challenge. This can be attributed to the larger volume of visual training data and a visual-centric bias in video-audio joint datasets. Moreover, Language Dominance reveals the impact of LLM decoders, with steep declines in HR from PA scores, as seen in FAVOR dropping from $92.0$ to $18.5$. This indicates a strong reliance on language priors, suggesting a need to better balance multi-modal integration.

**Response Tendencies of LMMs.** Certain models show atypical response patterns. GroundingGPT tends to answer "yes" carelessly, leading to high PA but low HR scores (e.g., $0$ in AL correlations). This suggests overconfidence or excessive human alignment during training, as also previously noted by other studies [31]. In contrast, Reka-core and VideoLLaMA2 exhibit cautious tendencies, showing higher HR than PA in many cases and occasionally very low PA scores (e.g., Reka-core's $25.0$ PA in AL correlations). This likely reflects safety alignment strategies to reject uncertain inputs with "no" responses. These contrasting response tendencies underscore the varied behavioral patterns in LMMs and highlight the need for more balanced training strategies that ensure accurate, context-dependent responses without overconfidence or excessive caution.

### 4.2.2 Analyzing Visual-only and Audio-only LMMs

Visual and audio-only LMMs show superior perception accuracy in their respective domains compared to Visual-Audio LMMs, as evidenced by higher PA scores in Tab. 3a and Tab. 3b. However, this advantage does not extend to mitigating hallucinations. Similar to Visual-Audio LMMs, single-modality models remain vulnerable to hallucinations caused by spurious inter-modality correlations. Despite previous efforts to address VL correlations, some models still exhibit poor HR scores, such as CogVLM2-Video, which scores $44$. Furthermore, AL correlations pose even greater challenges, with audio-only LMMs scoring between $30$ and $60$ in HR, underscoring the insufficient mitigation of hallucinations in audio-text interactions, likely due to the limited attention this issue has received in prior research [40]. Additionally, most Visual-only LMMs exhibit low HR scores for Language Dominance, hovering around $50$. This indicates a strong reliance on language priors, leading to hallucinations when visual input conflicts with linguistic expectations. These findings further strengthen our identification of the two key factors driving hallucinations.

### 4.3 Discussions

**Effects of Probing Granularities**. Our benchmark includes both object-level and event-level probing questions across subcategories[8]. As shown in Tab. 4, most models show lower PA scores for event-level queries than object-level ones, highlighting the challenge posed by temporal complexity and the limited availability of event-oriented training data. For Visual-Language (VL) spurious correlations, event-level probing yields higher HR scores than object-level probing. This may be due to the scarcity of event-level annotations in visual-text pretraining data, while object-level annotations are more prevalent, fostering stronger spurious correlations. Conversely, within Language Dominance under

---

[8]Audio-related subcategories exclusively contain event-level queries due to their temporal nature.

Table 4: Effects of Probing Modalities.

| Model | Visual Prob | | Audio Prob | |
|---|---|---|---|---|
| | pa | hr | pa | hr |
| Reka-Core | 96.6 | 86.7 | 57.1 | 83.5 |
| Gemini-1.5-Flash | 94.0 | 92.0 | 83.0 | 49.0 |
| Gemini-1.5-Pro | 92.0 | 90.0 | 80.0 | 44.0 |
| FAVOR | 94.0 | 85.0 | 95.0 | 53.0 |
| GroundingGPT | 96.0 | 35.0 | 99.0 | 1.0 |
| VideoLLaMA2 | 84.0 | 99.0 | 72.0 | 97.0 |

Table 5: Effects of LLM Sizes.

| Model | LLM Size | VL Cor | | Lan Dom | |
|---|---|---|---|---|---|
| | | pa | hr | pa | hr |
| PLLaVA | 7B | 89.5 | 93.0 | 75.0 | 52.0 |
| | 13B | 86.5 | 96.5 | 75.5 | 65.0 |
| | 34B | 91.0 | 94.5 | 75.5 | 74.0 |
| LLaVA-OV | 0.5B | 96.5 | 91.5 | 81.0 | 55.0 |
| | 7B | 94.0 | 88.0 | 87.5 | 69.5 |
| | 72B | 84.5 | 93.5 | 89.5 | 75.5 |

Unimodal over-reliance, HR scores are lower for event-level queries. This pattern is likely due to the autoregressive nature of large language models, which increases reliance on language priors as the length of processed sequences grows, heightening the risk of hallucinations, especially when longer event-related common-sense knowledge is involved.

**Effects of Probing Modalities**. The Visual-Audio-Language (VAL) subcategory examines spurious correlations arising from the co-occurrence of visual objects and audio events. It includes two probing types: (1) object-level queries about non-existent visual objects when frequently co-occurring audio events are present, and (2) event-level queries about non-existent audio events when frequently co-occurring visual objects are present. Despite both probing types originating from similar co-occurrence patterns, HR scores for event-level (audio) probing are significantly lower than those for object-level (visual) probing across all models (Tab. 4). This finding aligns with our analysis of Visual and Audio Dominance under Unimodal over-reliance, suggesting a bias towards visual data due to its abundance in training and the visual-centric nature of joint visual-audio pretraining. As a result, models tend to over-rely on visual cues, leading to more pronounced hallucinations when predicting non-existent audio events.

**Effects of LLM Sizes**. We analyzed the impact of LLM decoder sizes on two LMMs, PLLaVA and LLaVA-OneVision. As shown in Tab. 5, increasing the LLM size has minimal influence on HR scores for Visual-Language spurious correlations, supporting our claim that these correlations primarily arise from global appearance and co-occurrence patterns in training data. In contrast, larger LLM sizes consistently improve HR scores for Language Dominance. For example, LLaVA-OneVision's HR score increases from $55.0$ ($0.5B$ LLM) to $75.5$ ($34B$ LLM), suggesting that larger LLMs are more adept at managing complex or contradictory multi-modal inputs. Smaller LLMs, however, are more susceptible to overfitting to linguistic priors, leading to higher hallucination rates when faced with content that deviates from expected patterns.

Due to space constraints, potential future directions and additional analyses—including the effects of focus prompting, QA formatting, and question templates—are provided in Appendix D and B.

# 5  Related Works

**Large Multi-modal Models**. Recent LMMs like LLaVA [36] and Flamingo [3] utilize transformer architectures to enhance cross-modal understanding, enabling nuanced visual-text comprehension for tasks such as visual question answering and image-based dialogue. Beyond static image-text tasks, recent approaches have aimed to extend multi-modal capabilities by incorporating additional modalities like video and audio [11, 58, 13, 1, 45, 68, 64, 17], fostering richer context and enhancing the model's ability to handle a diverse range of multi-modal scenarios.

**Hallucinations in LMMs**. Hallucination, particularly object hallucination, has been extensively studied in LMMs that process image and text. This phenomenon arises when a model generates content inconsistent with the actual objects present in the input image. Various benchmarks have been developed to assess hallucination in vision-language tasks [31, 54, 19, 39, 9, 65, 67, 37], and several mitigation techniques have been proposed [4, 27, 22, 66, 49, 60]. However, research on hallucinations in LMMs beyond image-text tasks remains limited, with relatively few studies addressing hallucinations involving additional modalities such as audio and video [59, 40]. A concurrent effort, AVHBench [50], investigates unimodal over-reliance in audio-visual settings but does not address inter-modality correlation effects.

# 6 Conclusions

To the best of our knowledge, this paper is the first to systematically investigate and verify the two key contributors to hallucinations in Large Multi-modal Models (LMMs) across language, visual, and audio modalities: *over-reliance on unimodal priors* and *spurious inter-modality correlations*. We introduce The **C**urse of **M**ulti-**M**odality (**CMM**) benchmark, which features nuanced subcategories and granularities along with diagnostic metrics, enabling precise diagnosis of model limitations and guiding targeted improvements. By benchmarking various LMMs across diverse multi-modal contexts, we identified key vulnerabilities in current models, such as unbalanced multi-modal integration and biases arising from pretraining datasets. Our analyses provide fundamental insights into multi-modal learning, highlighting the need for improved alignment across multi-modal inputs and offering foundational guidance for developing more robust and reliable LMMs.

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

## A    Experimental Details

### A.1    Qualitative Demonstrations

For the demonstrations in the main paper Fig.2, we use three advanced LMMs capable of processing both visual and audio inputs. Gemini-1.5-pro [52], FAVOR-13B [46], and VideoLLaMA2-7B [11]. The case studies presented in the main paper Fig.2 analyze hallucination tendencies by computing $p_\theta($"yes"/"no" $\mid v, a', x)$ and $p_\theta($"yes"/"no" $\mid v', a, x)$, using VideoLLaMA2-7B as a representative model.

Furthermore, we extend our experiments from Fig.2 of the main paper, to justify existing tri-modality models over-rely on uni-modal priors to response [31, 27]. To achieve this, we sampled 20 failure cases with questions asking non-existent objects/events for each open-sourced LMM under the over-rely on visual and audio priors in CMM subsets (where the LMM's output probability on those test cases , but the ground truth answer is "no"). As shown in Fig. 4, we plotted the average along with standard deviations across blur/noise steps applied to specific input modalities. The observed trends align with our original findings in Figure 2, demonstrating that reducing information from the dominant modality forces the model to rely more on the targeted modality, effectively decreasing hallucination rates. This highlights the critical impact of modality dominance on hallucinations and underscores the necessity of robust cross-modal integration, particularly in scenarios involving mutual exclusion.

### A.2    Quantitative Validation

For the quantitative validation shown in main paper Fig.3, we curate 200 samples for each subcategory of hallucination.

**Visual-Language Experiments**. Each sample is a video-only raw file associated with a probing question targeting the existence of a non-existent object, while a frequently co-occurring object is present. The co-occurrence scores are computed from the WebVid-10M dataset [6], from which the video samples are also sourced. For instance, a video containing a bird is queried with "Did you see trees in the video?" since "bird" and "tree" frequently co-occur in the pretraining data, although no tree is visually present in the sample.

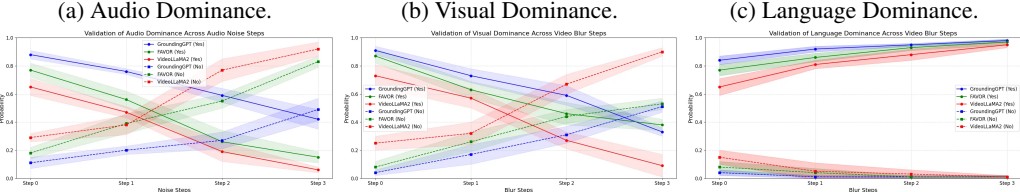

| (a) Audio Dominance. | (b) Visual Dominance. | (c) Language Dominance. |

Figure 4: Validation experiments on overreliance on unimodal priors.

**Audio-Language Experiments**. Given the temporal nature of audio, all queries are event-level. Each audio-only raw file is associated with a question about a non-existent audio event, while the subject of a related event can be recognized. For example, a dog whimpering is queried with "Did you hear dog barking?" Co-occurrence scores are computed from the audio-text pretraining dataset Auto-acd [48], which also provides the audio samples.

**Audio-Visual Experiments**. The co-occurrence scores are derived from the video-audio dataset AudioCaps [24], containing video samples with corresponding audio tracks. Each sample is queried about a non-existent visual object with a frequently co-occurring audio event, or vice versa.

The above experiments are conducted on three open-source LMMs that support both visual and audio inputs: FAVOR-13B [46], GroundingGPT-7B [32], and VideoLLaMA2-7B [11]. The frequencies displayed in the main paper Fig.3 represent the aggregated results across all three models.

# B  Additional Experiments

We provide additional experiments in this section.

## B.1  Effects of Focus Prompting

Table 6: Effects of prompting LMMs to focus on the key modality.

| Model | Visual Dom | | Audio Dom | |
|---|---|---|---|---|
| | *pa* | *hr* | *pa* | *hr* |
| Gemini-1.5-Flash | 99.5 | 44.0 | 98.0 | 5.0 |
| + *focus* | 97.0 | 66.0 | 88.0 | 34.5 |
| GroundingGPT | 99.0 | 73.0 | 94.5 | 46.5 |
| + *focus* | 89.5 | 93.0 | 75.0 | 52.0 |
| FAVOR | 87.5 | 85.5 | 79.5 | 58.0 |
| + *focus* | 94.0 | 88.0 | 87.5 | 69.5 |
| VideoLLaMA2 | 87.5 | 95.5 | 83.0 | 84.0 |
| + *focus* | 87.5 | 95.5 | 83.0 | 84.0 |

We conduct experiments prompting LMMs to focus on a single modality. Specifically, in scenarios where LMMs tend to rely on audio while ignoring visual inputs, we prompt them to focus more on the visual content. A similar approach is applied when LMMs over-rely on visual inputs while ignoring audio. As shown in Tab. 6, such prompting can reduce hallucinations to some extent. However, the improvements are not consistent across models and metrics, suggesting challenges in addressing hallucinations among tri-modalities. Furthermore, this approach presupposes prior knowledge of the decisive input modality, which is impractical in real-world scenarios.

## B.2  Effects of QA formatting

To avoid the potential influence of yes-no tendency [35], where LMMs tend to answer with *yes*, we reformulate our questions with multiple-choice formats, as demonstrated in Tab. 7. Such modifications ensures that LMMs cannot simply rely on yes-no tendencies to answer. CMM results with reformated questions are summarized in Tab. 8. Such results indicate that a leading proprietary LMM like Gemini-1.5-Flash suffers less from yes-no

Table 7: Example of different QA formatting.

| Yes or No Formatting | |
|---|---|
| Question | Did you hear bird chirping in the audio? Please answer with yes or no. |
| Ground Truth | No. |
| A or B Formatting | |
| Question | Did you hear bird chirping in the audio? A. Yes. B. No. Select the best option for the question. |
| Ground Truth | B. |

tendency. This also suggests that CMM poses challenges in robustness to existing models in tri-modality scenarios, which is less attended to in previous hallucination evaluations [31, 39, 19, 67].

Table 8: Effects of QA formatting. FAVOR and GroundingGPT do not follow instructions to response with A or B, thus not included.

| Model | Spurious Correlation | | Unimodal Overreliance | |
|---|---|---|---|---|
| | *pa* | *hr* | *pa* | *hr* |
| Gemini-1.5-Flash | 90.2 | 66.7 | 86.7 | 61.7 |
| *w. A or B* | 94.0 | 60.8 | 94.7 | 52.1 |
| Video-LLaMA2 | 76.8 | 92.7 | 66.5 | 69.5 |
| *w. A or B* | 94.7 | 44.5 | 94.0 | 25.2 |

## B.3 Effect of Question Template

Table 9: Question Template Variations.

| Template 1 | |
|---|---|
| Q | Did you hear bird chirping in the audio? |
| A | No. |
| Template 2 | |
| Q | Can you hear bird chirping in the audio? |
| A | No. |
| Template 3 | |
| Q | Is bird chirping audible in the audio? |
| A | No. |

To evaluate the potential influence of question template variations on results, we ask six LMMs with three prompt variations, as listed in Tab. 9. The quantitative results are summarized in Tab. 10. As the table indicates, most existing models are drastically affected by question templates variations, suggesting challenges in tri-modality alignment. We hope such results inspire future designs that suffer less from these issues.

## B.4 Evaluation of GPT-Series Models

As the GPT-series models do not support audio inputs, we evaluated three flagship models (GPT-4.1, GPT-4.1 mini, and GPT-4.1 nano) on the visual–language (VL) subsets of our CMM benchmark.

Across all models, hallucination resistance (*HR*) is consistently lower for event-level queries than for object-level ones, reflecting the difficulty of overcoming temporal or event-based language priors. The vulnerability becomes

Table 10: Effects of Question Template.

| | Spurious Inter-modality Correlation | | | | | | Uni-modality Over-reliance | | | | | | Overall | |
|---|---|---|---|---|---|---|---|---|---|---|---|---|---|---|
| | VL | | AL | | VAL | | V | | A | | L | | | |
| | pa | hr | pa | hr | pa | hr | pa | hr | pa | hr | pa | hr | pa | hr |
| Gemini-1.5-Flash | | | | | | | | | | | | | | |
| + prompt 1 | 93.5 | 90.0 | 88.5 | 39.5 | 88.5 | 70.5 | 79.0 | 36.5 | 90.5 | 86.5 | 90.5 | 62.0 | 88.4 | 64.2 |
| + prompt 2 | 96.0 | 82.5 | 93.0 | 38.5 | 97.5 | 51.8 | 97.0 | 23.5 | 95.5 | 72.0 | 94.0 | 59.5 | 95.5 | 54.6 |
| + prompt 3 | 96.5 | 81.0 | 87.0 | 53.5 | 97.5 | 59.0 | 95.5 | 20.0 | 96.0 | 71.0 | 97.5 | 45.0 | 95.0 | 54.9 |
| Δ | 3.0 | 9.0 | 6.0 | 14.0 | 9.0 | 18.7 | 18.0 | 16.5 | 5.5 | 15.5 | 7.0 | 17.0 | 7.1 | 9.6 |
| GroundingGPT | | | | | | | | | | | | | | |
| + prompt 1 | 95.5 | 36.5 | 100.0 | 0.0 | 97.5 | 18.0 | 99.5 | 1.0 | 98.5 | 23.5 | 88.5 | 7.0 | 96.6 | 14.3 |
| + prompt 2 | 97.0 | 46.5 | 100.0 | 0.0 | 97.5 | 28.5 | 97.0 | 0.5 | 96.0 | 38.5 | 90.5 | 14.0 | 96.3 | 21.3 |
| + prompt 3 | 97.5 | 42.0 | 99.5 | 0.0 | 94.5 | 52.0 | 82.0 | 2.5 | 96.5 | 43.5 | 91.0 | 13.5 | 93.5 | 25.6 |
| Δ | 2.0 | 10.0 | 0.5 | 0.0 | 3.0 | 34.0 | 17.5 | 2.0 | 2.5 | 20.0 | 2.5 | 7.0 | 3.1 | 11.3 |
| PandaGPT | | | | | | | | | | | | | | |
| + prompt 1 | 96.5 | 27.0 | 90.5 | 11.0 | 84.5 | 17.5 | 89.0 | 13.5 | 95.0 | 17.5 | 87.0 | 18.5 | 90.5 | 17.5 |
| + prompt 2 | 99.0 | 8.5 | 94.5 | 0.0 | 95.5 | 7.5 | 94.5 | 4.0 | 97.0 | 7.5 | 98.0 | 1.5 | 96.6 | 4.8 |
| + prompt 3 | 98.5 | 2.0 | 94.5 | 0.5 | 96.0 | 1.0 | 98.0 | 4.5 | 99.0 | 0.0 | 98.5 | 0.0 | 97.5 | 1.3 |
| Δ | 2.5 | 25.0 | 4.0 | 11.0 | 11.5 | 16.5 | 9.0 | 9.5 | 4.0 | 17.5 | 11.5 | 188.5 | 7.0 | 16.2 |
| FAVOR | | | | | | | | | | | | | | |
| + prompt 1 | 91.0 | 55.0 | 94.5 | 45.0 | 94.5 | 69.0 | 89.0 | 21.5 | 92.0 | 43.5 | 92.0 | 18.5 | 92.2 | 42.1 |
| + prompt 2 | 92.5 | 54.5 | 91.5 | 58.5 | 91.0 | 77.5 | 91.5 | 25.5 | 93.0 | 56.0 | 89.0 | 20.5 | 91.4 | 48.8 |
| + prompt 3 | 98.5 | 32.0 | 96.0 | 43.0 | 95.0 | 58.0 | 87.0 | 23.5 | 89.0 | 47.5 | 88.0 | 23.5 | 92.3 | 37.9 |
| Δ | 7.5 | 23.0 | 4.5 | 15.5 | 4.0 | 19.5 | 4.5 | 4.0 | 4.0 | 12.5 | 4.0 | 5.0 | 0.9 | 10.9 |
| Video-Salmonn | | | | | | | | | | | | | | |
| + prompt 1 | 60.0 | 71.5 | 70.0 | 89.0 | 67.0 | 80.0 | 59.5 | 90.0 | 61.0 | 51.5 | 59.0 | 30.5 | 62.8 | 68.8 |
| + prompt 2 | 67.5 | 63.0 | 82.5 | 76.5 | 73.0 | 80.5 | 78.5 | 66.0 | 56.5 | 57.0 | 64.5 | 20.0 | 70.4 | 60.5 |
| + prompt 3 | 81.5 | 29.5 | 82.5 | 63.0 | 78.0 | 73.0 | 81.5 | 54.5 | 74.0 | 39.0 | 67.5 | 17.0 | 77.5 | 46.0 |
| Δ | 21.5 | 42.0 | 12.5 | 26.0 | 11.0 | 7.5 | 22.0 | 35.5 | 17.5 | 18.0 | 8.5 | 13.5 | 14.7 | 22.8 |
| VideoLLaMA2 | | | | | | | | | | | | | | |
| + prompt 1 | 75.0 | 86.0 | 77.5 | 94.0 | 78.0 | 98.0 | 62.0 | 75.5 | 80.0 | 90.0 | 57.5 | 43.0 | 71.7 | 81.1 |
| + prompt 2 | 89.0 | 90.5 | 81.0 | 89.5 | 86.5 | 96.5 | 71.0 | 64.0 | 82.0 | 92.0 | 83.5 | 25.5 | 82.2 | 76.3 |
| + prompt 3 | 82.5 | 91.0 | 82.0 | 86.5 | 86.5 | 93.5 | 72.0 | 62.0 | 83.0 | 90.0 | 70.5 | 47.5 | 79.4 | 78.4 |
| Δ | 14.0 | 5.0 | 4.5 | 7.5 | 8.5 | 4.5 | 10.0 | 13.5 | 3.0 | 2.0 | 26.0 | 17.5 | 10.5 | 4.8 |

| Model | Granulariy | PA | HR |
|---|---|---|---|
| GPT-4.1 | Object | 97.0 | 93.9 |
| | Event | 85.0 | 96.0 |
| GPT-4.1 mini | Object | 93.8 | 92.9 |
| | Event | 84.0 | 96.0 |
| GPT-4.1 nano | Object | 98.0 | 82.8 |
| | Event | 89.9 | 90.0 |

(a) VL Correlations subset.

| Model | Granularity | PA | HR |
|---|---|---|---|
| GPT-4.1 | Object | 90.0 | 94.0 |
| | Event | 79.0 | 81.0 |
| GPT-4.1 mini | Object | 85.0 | 91.0 |
| | Event | 68.0 | 76.0 |
| GPT-4.1 nano | Object | 81.0 | 63.0 |
| | Event | 75.3 | 35.4 |

(b) Language Dominance subset.

increasingly pronounced for smaller models: *HR* on event-level queries drops from 81.0 (GPT-4.1) to 35.4 (GPT-4.1 nano), demonstrating the sensitivity of lightweight architectures to language dominance.

## B.5 Chain-of-Thought (CoT) Reasoning Experiments

We further examined the effect of explicit Chain-of-Thought (CoT) prompting using the instruction: *"Think step-by-step and then answer with yes or no at the end."* Each cell in Table 12 reports Perception Accuracy (PA) / Hallucination Resistance (HR).

**Analysis.** CoT prompting introduces a clear trade-off between reasoning and perception. **Cautious reasoners** (Gemini models) tend to lose perception accuracy while gaining limited hallucination resistance, indicating interference between linguistic reasoning and visual grounding. **Over-confident reasoners** (PandaGPT) display the opposite pattern—stronger PA but catastrophic HR collapse—suggesting amplification of pre-existing biases.

Table 12: Original vs. Chain-of-Thought (CoT) performance comparison across benchmark subsets.

| Model | Setting | VL Corr. | AL Corr. | VAL Corr. | Visual Dom. | Audio Dom. | Lang. Dom. |
|---|---|---|---|---|---|---|---|
| Gemini-1.5 Pro | Original | 91.0 / 90.5 | 94.0 / 14.5 | 86.0 / 67.0 | 82.5 / 34.0 | 90.5 / 82.0 | 78.5 / 61.5 |
| | CoT | 70.5 / 81.0 | 70.0 / 69.0 | 70.5 / 55.5 | 55.0 / 32.0 | 71.5 / 65.0 | 56.0 / 66.0 |
| Gemini-1.5 Flash | Original | 93.5 / 90.0 | 88.5 / 39.5 | 88.5 / 70.5 | 79.0 / 36.5 | 90.5 / 86.5 | 90.5 / 62.0 |
| | CoT | 86.5 / 90.0 | 72.0 / 69.0 | 82.5 / 77.0 | 76.5 / 43.5 | 88.0 / 80.5 | 84.0 / 74.5 |
| Gemini-2.0 Flash | Original | 95.0 / 83.5 | 98.5 / 47.0 | 97.5 / 68.0 | 96.5 / 36.0 | 93.0 / 71.0 | 94.0 / 62.5 |
| | CoT | 87.5 / 92.0 | 92.0 / 53.5 | 92.0 / 76.5 | 88.5 / 38.5 | 83.0 / 78.5 | 86.5 / 66.0 |
| Qwen2.5-Omni | Original | 88.5 / 97.0 | 92.0 / 83.5 | 91.0 / 97.5 | 89.5 / 74.5 | 82.5 / 85.0 | 68.5 / 85.0 |
| | CoT | 89.0 / 97.5 | 92.5 / 82.0 | 87.0 / 97.5 | 86.0 / 79.0 | 77.0 / 85.5 | 66.5 / 83.0 |
| PandaGPT | Original | 96.5 / 27.0 | 90.5 / 11.0 | 84.5 / 17.5 | 89.0 / 13.5 | 95.0 / 17.5 | 87.0 / 18.5 |
| | CoT | 98.0 / 15.0 | 95.0 / 2.0 | 96.0 / 3.0 | 97.0 / 4.0 | 99.0 / 2.0 | 96.0 / 3.0 |
| Video-Salmonn | Original | 60.0 / 71.5 | 70.0 / 89.0 | 67.0 / 80.0 | 59.5 / 90.0 | 61.0 / 51.5 | 59.0 / 30.5 |
| | CoT | 60.0 / 62.0 | 71.5 / 85.5 | 63.5 / 84.5 | 59.0 / 85.5 | 55.0 / 57.0 | 53.5 / 37.5 |

**Mixed responders** (QwenOmni and Video-Salmonn) exhibit minimal or inconsistent changes. These results show that longer reasoning chains may intensify reliance on language priors and weaken perceptual grounding.

# C  Data Details

## C.1  Data Statistics

**Length Distribution**. We summarize the audio and video length distributions of CMM in Fig. 5. We summarize three types of input data involved in CMM, including (a) 400 audio-only samples, (b) 800 video-only samples, and (c) remaining 1200 paired audio-visual samples. The audio-only data are extracted from the large-scale Auto-ACD, where most samples contain approximately 10 seconds of audio, and the length distribution of our audio-only samples follows the characteristics.

(a) Distribution of CMM Audio Lengths.  (b) Distribution of CMM Video Lengths.  (c) Distribution of Audio-Visual Lengths.

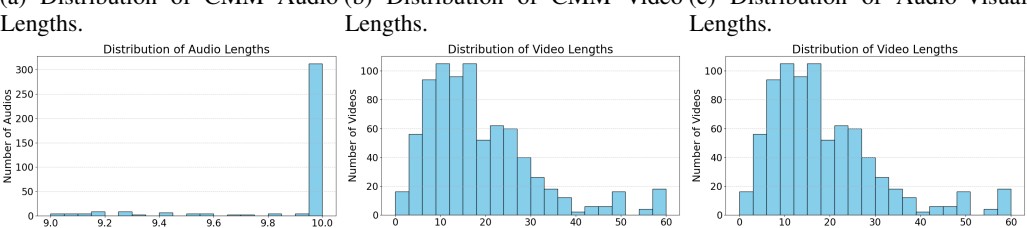

Figure 5: Length Distribution.

**Object/Event Statistics**. We provide the statistics of objects and events in detail. Specifically, we include the most frequent 10 existent and non-existent objects, visual events and audio events in Fig. 6. The full distributions of object, visual event and audio event frequency are summarized in Fig. 7, Fig. 8, and Fig. 9, respectively. It should be pointed out that our aim is not to replicate natural distribution with our benchmark. Instead, we highlight tri-modal test cases that challenge existing LMMs with spurious inter-modality spurious correlations and over-reliance on uni-modal priors, whereby we hope inspiring more robust and safe models that suffer less from hallucinations.

## C.2  Benchmark Data Construction Details

The benchmark is designed to evaluate hallucination scenarios across multiple modalities, targeting specific LMM tendencies such as Over Reliance on individual modalities and spurious inter-modality correlations. It comprises video, audio, and textual inputs with probing questions aimed at assessing the presence or absence of objects or events in these modalities. Precise annotation is employed to ensure a thorough evaluation of LMM performance in multimodal contexts.

### C.2.1  Over Reliance on Unimodal Priors

To assess how LMMs may excessively depend on a single modality (visual, audio, or language), we construct targeted probing queries that test this Over Reliance while potentially neglecting complementary information.

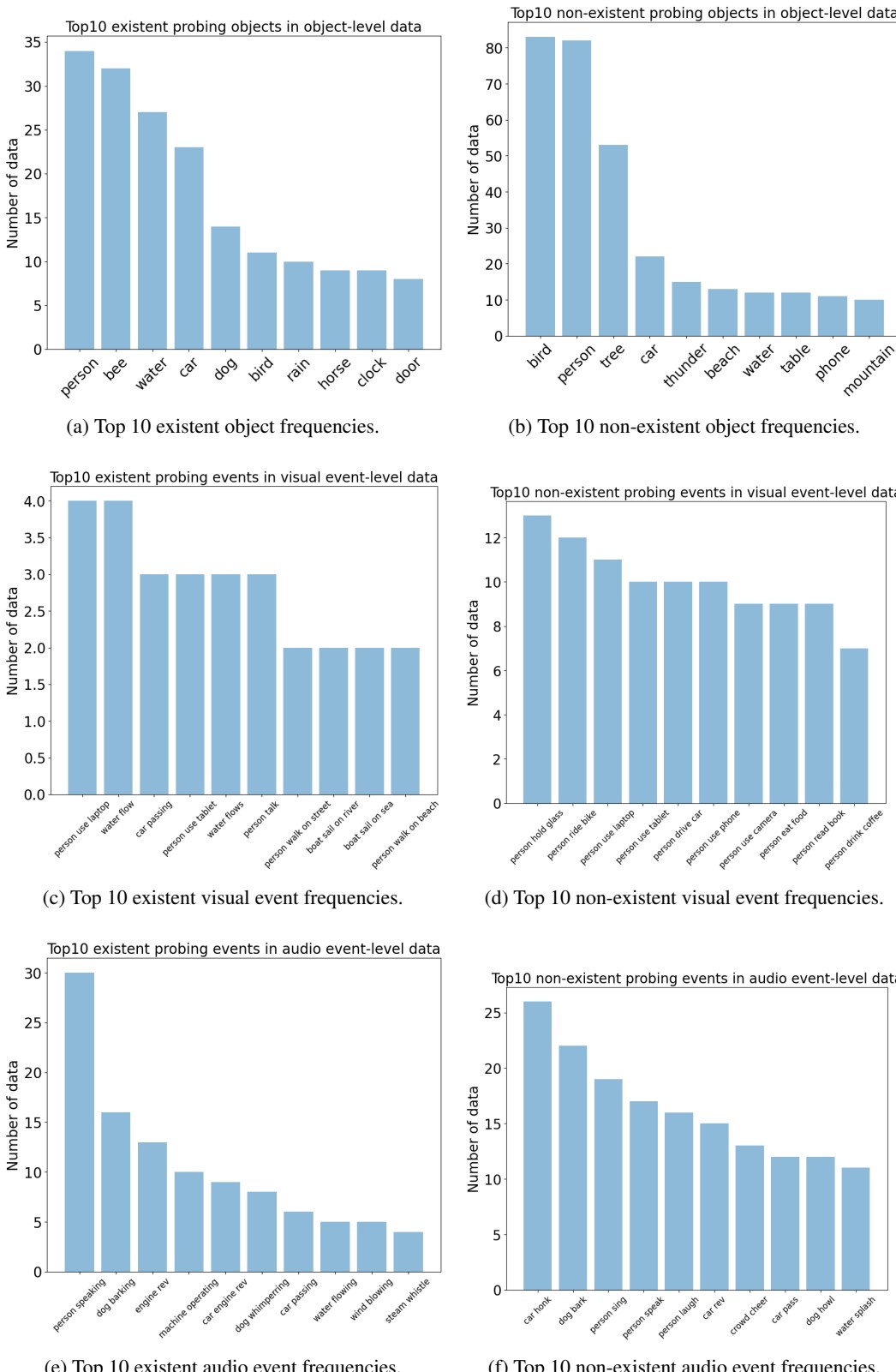

(a) Top 10 existent object frequencies.

(b) Top 10 non-existent object frequencies.

(c) Top 10 existent visual event frequencies.

(d) Top 10 non-existent visual event frequencies.

(e) Top 10 existent audio event frequencies.

(f) Top 10 non-existent audio event frequencies.

Figure 6: Statistics of object and event frequencies in our probing questions.

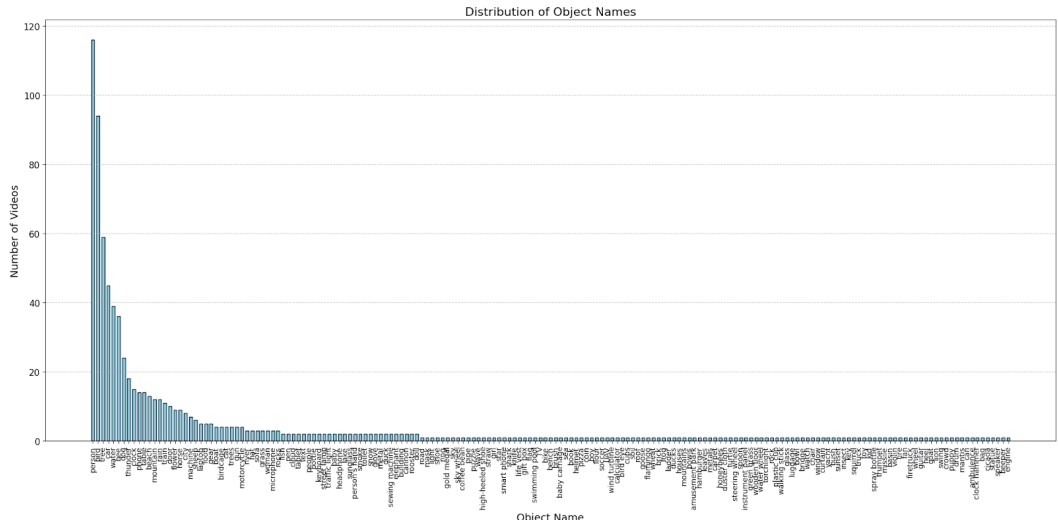

Figure 7: Distribution of Object Names.

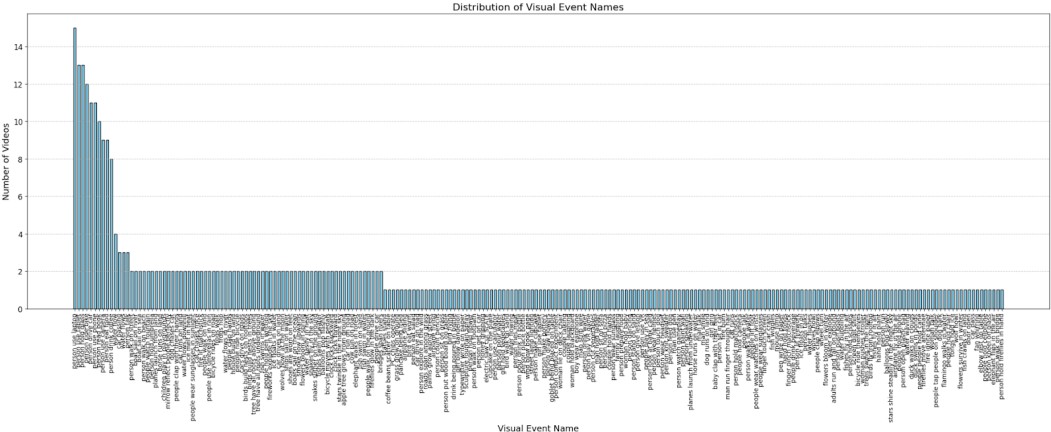

Figure 8: Distribution of Visual Event Names.

**Visual Dominance**. The Visual Dominance subcategory examines the extent to which LMMs over-rely on visual content, potentially leading to hallucinated sound events that are often associated with visual objects. All probing questions focus on audio events. For queries about existent sound events, the ground truth "yes" is derived from direct human annotation. To identify non-existent sound events, we use the AudioCaps dataset [24], which provides short captions describing the audio track. Objects associated with these audio events are extracted using LLaMA3 [16] from the audio caption, while visual objects are identified from video frames using InternVL2 [10]. Samples where visual objects do not correspond to any audio content are filtered and manually verified, with the ground truth set to "no." All raw video-audio pairs are sourced from AudioCaps.

**Audio Dominance**. The Audio Dominance subcategory explores how LMMs may over-rely on audio cues, leading to hallucinations of visual content. Here, questions probe the presence of visual objects. For existent objects, the ground truth "yes" is annotated manually. To find non-existent objects, we filter samples where the objects indicated by audio cues are not visually present in the video. These samples undergo manual review to ensure accurate annotation, with the ground truth as "no." All raw video-audio pairs are also sourced from AudioCaps.

**Language Dominance**. The Language Dominance subcategory targets hallucinations caused by the LMMs' dependence on language priors from pretraining corpora. This category focuses on common-sense events and object attributes. We manually define sets of typical events (e.g., "fish swim in water") and object characteristics (e.g., "yellow banana"). Videos depicting anti-common-sense scenarios (e.g., "fish fly in the air," "black banana") are then collected from YouTube. For queries probing existent content, the ground truth "yes" corresponds to the

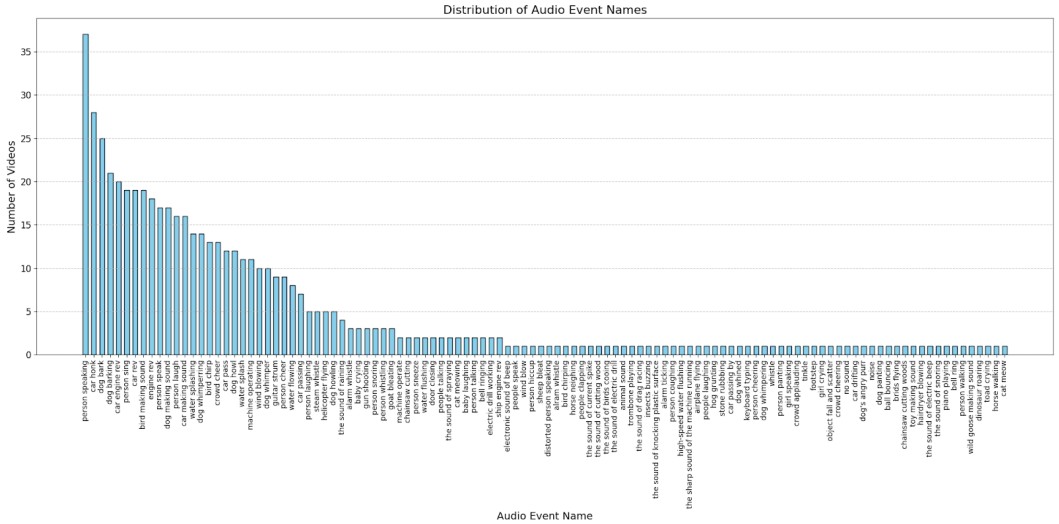

Figure 9: Distribution of Audio Event Names.

anti-common-sense object/event depicted in the video. Conversely, non-existent content queries, which are the common-sense versions that do not match the video, have the ground truth "no."

Each subcategory includes 200 video-audio or video-only samples, each accompanied by two probing questions: one querying an existent object/event ("yes"), and another probing a non-existent one ("no"). For subcategories containing both object- and event-level probing, the dataset is balanced with equal numbers of object- and event-level queries.

### C.2.2 Spurious Inter-modality Correlations

This section outlines the construction of queries targeting *Spurious Inter-modality Correlations*, where hallucinations arise from misleading associations between different modalities learned during pretraining. These correlations are probed at both object- and event-level granularities.

**Visual-Language**. This occurs when LMMs hallucinate visual objects due to associations learned from patterns in video-caption pretraining data. Queries in this subcategory are developed based on two factors: global appearance frequencies and co-occurrence patterns within the data.

*Object-level* queries are derived from two sources: (i) global appearance frequencies, where the model is asked about frequent objects that are absent in the video (e.g., "Did you see a tree in the video?" when no tree is present), and (ii) co-occurrence patterns, where queries target non-existent objects that are often seen alongside other objects in the pretraining data (e.g., "Did you see a phone in the video?" when a human is present but no phone).

*Event-level* queries similarly explore global appearance frequencies by probing events that frequently occur in pretraining data but are not present in the video. For co-occurrence patterns, event-level queries are designed around subject-fixed action-object pairs, such as "Did you see a person using a phone in the video?" when the person is engaged in a different action like walking.

Both global frequencies and co-occurrence data are extracted from the large-scale video-caption pretraining dataset WebVid10M. Probing samples are curated accordingly from the same source.

**Audio-Language**. This subcategory assesses correlations learned from audio-caption pretraining, leading to potential hallucinations of audio events based on their appearance or co-occurrence in the training data. Due to the temporal nature of audio, all queries are event-level.

*Event-level* queries focus on global appearance frequencies, probing for hallucinated audio events that are common in the pretraining data but absent from the audio track (e.g., "Did you hear a dog barking?" when no such sound exists). Co-occurrence queries involve subject-fixed action-object pairs, targeting frequently co-occurring events (e.g., "Did you hear a dog barking?" when only dog whimpering is present).

The dataset Auto-acd is used for constructing these queries, ensuring a balanced representation of global appearance and co-occurrence-based patterns.

**Visual-Audio-Language**. The Visual-Audio-Language subcategory captures cross-modal hallucinations, where visual objects are hallucinated based on audio cues, and vice versa.

*Object-level* queries target visual objects that are hallucinated based on associated sound events (e.g., "Did you see a tree in the video?" when bird chirping is present without any tree visible).

*Event-level* queries test for non-existent audio events that are frequently co-occurred with visual objects in training data (e.g., "Did you hear car revving?" when a human is visible without any car sound).

The co-occurrence frequencies between visual objects and audio events are computed using the Auto-ACD dataset, with the visual and audio content manually checked and annotated by human reviewers. Queries are evenly split between probing audio events and visual objects.

For all subcategories, there is a balance between object-level and event-level queries. Additionally, the samples constructed from global appearance frequencies and co-occurrence patterns are evenly distributed.

### C.3 Frequent Patterns in Pretraining Datasets

The following outlines the frequent global appearances and co-occurrence patterns derived from major pretraining datasets, which is used to construct our benchmark. These patterns reflect common associations across modalities, contributing to spurious correlations within LMMs during pretraining.

---

**Patterns in Pretraining Datasets**

**Visual-Language Correlations from WebVid-10M**

- *Object-level*
    - **Top appeared objects**: [beach, boat, car, city, flower, mountain, person, phone, tree, water]
    - **Top co-occurrences**: [beach-person, car-person, city-person, dog-person, food-person, laptop-person, mountain-person, phone-person, tree-person, water-person]
- *Event-level*
    - **Top appeared events**: [person drinks coffee, person drives car, person eats food, person holds glass, person reads book, person rides bike, person uses camera, person uses laptop, person uses phone, person uses tablet]
    - **Top co-occurred (subject)-(action object) pairs**: [person-drinks coffee, person-drives car, person-eats food, person-holds glass, person-reads book, person-rides bike, person-uses camera, person-uses laptop, person-uses phone, person-uses tablet]

**Audio-Language Correlations from Auto-acd**

- *Event-level (since audio is inherently temporal)*
    - **Top appeared events**: [bird chirps, car passes, car revs, crowd cheers, dog barks, guitar strums, person laughs, person sings, person speaks, water splashes]
    - **Top co-occurred (subject)-(action object) pairs**: [car-honks, car-passes, car-revs, dog-barks, dog-howls, dog-whimpers, person-cheers, person-laughs, person-sings, person-speaks]

**Visual-Audio-Language from AudioCaps**

- *Cross-modality (visual object)-(audio event) co-occurrences*
    - **Top co-occurrences**: [person-bird chirping, tree-bird chirping, tree-car passing, person-dog barking, car-person speaking, table-person speaking, tree-person walking, person-water splashing, dog-person speaking, person-car revving, water-person speaking]

---

## D  Future Directions

Our analysis identifies key vulnerabilities in current LMMs, representing only a subset of broader challenges. These include but are not limited to unbalanced cross-modal integration, often with visual dominance overshadowing audio or text cues; spurious inter-modality correlations arising from training biases; overreliance on linguistic priors from large-scale LLM pretraining; and divergent response tendencies—either overconfident approval or overly cautious rejection. To address these challenges, we propose several potential directions for reference:

- Balanced Multi-modal Training Data. Creating datasets with balanced modality representation and diverse temporal annotations to reduce visual biases and improve event-level understanding.

- Advanced Cross-modal Fusion. Implementing dynamic fusion strategies to adjust modality importance based on context can improve multimodal integration and reduce hallucination.

- Mitigating Linguistic Priors. Fine-tuning LMMs with contextually diverse prompts and incorporating visual/audio fact-checking mechanisms can decrease overreliance on language priors.

- Refined Safety Alignment. Establishing balanced response strategies to avoid overconfidence or excessive caution ensures accurate interpretation, even for ambiguous inputs.

# E   Limitation

While our study introduces a structured benchmark to evaluate hallucinations in LMMs, several limitations remain. First, although our analysis identifies two key contributors—unimodal prior overreliance and spurious inter-modality correlations—these do not exhaustively capture all possible causes of hallucination. Other underexplored factors, such as modality misalignment due to temporal inconsistency or more complex and entangled scenarios, may also contribute and warrant further investigation.

Second, current open-source Visual-Audio-Language (VAL) models exhibit limited instruction-following capabilities and display a strong response bias toward affirmative answers. Despite our efforts to mitigate this via prompt formatting and binary answer constraints, this bias persists, which may confound evaluation and necessitate the development of more robust instruction-following capability.

Lastly, our benchmark focuses on short, binary probing questions. While effective for diagnosis, this format may not fully reflect the complexity of real-world multimodal tasks, where nuanced, multi-step reasoning and open-ended generation are required. Extending evaluations to cover such settings remains an important direction for future work.

# F   Computation Resource

For benchmark evaluation, the majority of experiments are conducted using open-source models ranging from 7B to 13B parameters. These models can be deployed on a single GPU with 40–80GB of memory, with inference times ranging from a few minutes to approximately 30 minutes, depending on the implementation efficiency of each model's codebase.

For experiments involving larger model sizes (34B and 72B) used in the analysis of LLM size effects, 2–4 80GB-GPUs are required to support inference. Evaluations of proprietary models are performed via their respective official APIs, as these models are not publicly available for local deployment.

# G   Broader Impacts

This work presents a systematic benchmark for evaluating hallucinations in large multimodal models (LMMs) across language, visual, and audio modalities. By identifying key failure modes and providing fine-grained diagnostics, our framework can facilitate the development of more reliable and robust multimodal systems. This has potential benefits in applications such as assistive technologies, education, and content moderation, where accurate multimodal understanding is critical.

While improved model reliability may accelerate deployment in real-world scenarios, we emphasize that our contribution is primarily diagnostic in nature. We release the benchmark and evaluation results to support transparency and reproducibility. We hope this work encourages continued research into model alignment and robustness, and informs best practices for safe and responsible development of LMMs.

# H   Safeguard

All data included in our benchmark have undergone rigorous manual verification to ensure safety and appropriateness for public research use. Each sample—whether sourced from licensed datasets such as AudioCaps and WebVid-10M or manually collected from YouTube—was individually reviewed to exclude any content involving identifiable individuals, private settings, or potentially harmful, sensitive, or inappropriate material.

In addition to verifying the multimodal inputs, we manually reviewed all associated probing questions to ensure they are neutral, non-offensive, and free from bias. This careful curation process reflects our commitment to responsible data sharing and minimizes risks related to misuse or unintended harms.

We encourage responsible use of the benchmark and provide clear documentation on its intended purpose: evaluating hallucination robustness in LMMs. Researchers are advised to use the benchmark solely for academic and diagnostic purposes and not for fine-tuning generative models without proper safety measures.

# I License for Existing Assets

The models we use to evaluate CMM in this paper, including proprietary models such as Gemini and Reka Core, open-source models like VideoSalmonn and VideoLLaMA 2, are under permissive license for academic purpose. For code with which we implement our evaluation, we use VLMEvalkit [15] for most models, which is with Apache-2.0 license and permissive to use.

