# OpenReview forum: "The Curse of Multi-Modalities: Evaluating Hallucinations of Large Multimodal Models across Language, Visual, and Audio"
_NeurIPS.cc/2025/Datasets_and_Benchmarks_Track — NeurIPS 2025 Datasets and Benchmarks Track poster_

### Official Review · Reviewer_w2gh · 2025-06-13

**Rating:** 5
**Confidence:** 3

**Summary:**

This paper investigates hallucinations in large multi-modal models (LMMs) involving language, vision, and audio. It identifies key causes—overreliance on unimodal priors and spurious cross-modal correlations—and introduces a benchmark, The Curse of Multi-Modalities (CMM), to systematically evaluate these issues. Findings reveal integration imbalances and training data biases, highlighting the need for better cross-modal learning and hallucination mitigation.

**Dataset Code Accessibility:**

Yes

**Ethical Considerations:**

No, there are no or only very minor ethics concerns

**Final Justification:**

Thank you for your response. The additional explanations have addressed my concerns satisfactorily. I will maintain my original rating.

**Limitations Weaknesses:**

The evaluation is conducted solely by having the model choose "yes" or "no," which lacks an assessment of the model's chain-of-thought. It is worth exploring whether guiding the model to think could alleviate the problem of excessive modality reliance.

**Strengths Contributions:**

1. This paper analyzes the causes of hallucinations in multimodal large models from the perspective of over-reliance on a certain modality. This approach is insightful and provides inspiration for future research.
2. The collected CMMBenchmark is capable of systematically evaluating hallucinations caused by inter-modality dependency, offering an effective evaluation method for hallucinations in large models.
3. The authors have elaborated on the collection process of the benchmark in detail and have optimized the QA formatting to avoid a prior bias towards "yes" or "no" answers. The processing is convincing.
4. In terms of experiments, the proposed benchmark was used to evaluate various large models, and the results are relatively complete.

---

> ### Author Rebuttal · Authors · 2025-07-30
>
> We are very grateful to the reviewer for your positive feedback and for this insightful suggestion. The original binary "yes/no" format was chosen for objective and scalable evaluation, but we agree that exploring the role of Chain-of-Thought (CoT) reasoning is a valuable and important direction.
>
> Inspired by your suggestion, we have conducted new experiments using a CoT prompt: `"Think step-by-step and then answer with yes or no at the end"`. We have evaluated three proprietary Gemini models and three open-source models.
>
> The detailed results below indicate that while CoT can be beneficial in some scenarios, it is **not a universal solution**. It can introduce **complex trade-offs**, often improving performance in one area while significantly degrading it in another.
>
> ### **Original vs. Chain-of-Thought (CoT) Performance Comparison**
> *Each cell shows Perception Accuracy (PA) / Hallucination Resistance (HR) Score.*
>
> | Model | Setting | VL Corr. | AL Corr. | VAL Corr. | Visual Dom. | Audio Dom. | Language Dom. |
> | :--- | :--- | :--- | :--- | :--- | :--- | :--- | :--- |
> | **Gemini-1.5 Pro** | Original  | 91.0 / 90.5 | 94.0 / 14.5 | 86.0 / 67.0 | 82.5 / 34.0 | 90.5 / 82.0 | 78.5 / 61.5 |
> | | CoT  | 70.5 / 81.0 | 70.0 / 69.0 | 70.5 / 55.5 | 55.0 / 32.0 | 71.5 / 65.0 | 56.0 / 66.0 |
> | **Gemini-1.5 Flash** | Original  | 93.5 / 90.0 | 88.5 / 39.5 | 88.5 / 70.5 | 79.0 / 36.5 | 90.5 / 86.5 | 90.5 / 62.0 |
> | | CoT  | 86.5 / 90.0 | 72.0 / 69.0 | 82.5 / 77.0 | 76.5 / 43.5 | 88.0 / 80.5 | 84.0 / 74.5 |
> | **Gemini-2.0 Flash** | Original  | 95.0 / 83.5 | 98.5 / 47.0 | 97.5 / 68.0 | 96.5 / 36.0 | 93.0 / 71.0 | 94.0 / 62.5 |
> | | CoT  | 87.5 / 92.0 | 92.0 / 53.5 | 92.0 / 76.5 | 88.5 / 38.5 | 83.0 / 78.5 | 86.5 / 66.0 |
> | **Qwen2.5-Omni** | Original  | 88.5 / 97.0 | 92.0 / 83.5 | 91.0 / 97.5 | 89.5 / 74.5 | 82.5 / 80.0 | 68.5 / 85.0 |
> | | CoT | 89.0 / 97.5 | 92.5 / 82.0 | 87.0 / 97.5 | 86.0 / 79.0 | 77.0 / 85.5 | 66.5 / 83.0 |
> | **PandaGPT** | Original  | 96.5 / 27.0 | 90.5 / 11.0 | 84.5 / 17.5 | 89.0 / 13.5 | 95.0 / 17.5 | 87.0 / 18.5 |
> | | CoT | 98.0 / 15.0 | 95.0 / 2.0 | 96.0 / 3.0 | 97.0 / 4.0 | 99.0 / 2.0 | 96.0 / 3.0 |
> | **Video-Salmonn** | Original  | 60.0 / 71.5 | 70.0 / 89.0 | 67.0 / 80.0 | 59.5 / 90.0 | 61.0 / 51.5 | 59.0 / 30.5 |
> | | CoT | 60.0 / 62.0 | 71.5 / 85.5 | 63.5 / 84.5 | 59.0 / 85.5 | 55.0 / 57.0 | 53.5 / 37.5 |
>
>
> The results from our Chain-of-Thought (CoT) experiments reveal a **complex and highly model-dependent landscape**. Far from being a universal solution, CoT prompting interacts with the intrinsic biases of different models in distinct ways, highlighting the value of our fine-grained benchmark. We observe three main behavioral patterns:
>
> 1.  **The "Cautious Reasoners" (Gemini Series):** These models exhibit a consistent trade-off. When prompted to reason, their Perception Accuracy (PA) for correctly identifying existing objects and events consistently degrades. For instance, Gemini-1.5 Pro's PA in the Visual Dominance category drops sharply from 82.5 to 55.0. In exchange, they sometimes achieve better Hallucination Resistance (HR), particularly in the Audio-Language Correlations category. This suggests that for these models, the reasoning process can **interfere with their ability to ground themselves in the primary multimodal evidence**.
>
> 2.  **The "Over-Confident Reasoner" (PandaGPT):** This model demonstrates a more dangerous failure mode. CoT prompting **dramatically improves** its PA across all categories, but causes a **catastrophic collapse in its HR**. For example, its HR against Audio-Language correlations plummets from 11.0 to just 2.0. This indicates that for PandaGPT, step-by-step reasoning amplifies a strong existing bias, causing it to confidently agree with almost any suggestion and hallucinate prolifically.
>
> 3.  **The "Mixed Responders" (QwenOmni, Video-Salmonn):** These models show less predictable behavior. QwenOmni’s performance remains relatively stable, while Video-Salmonn shows minor, inconsistent shifts. This highlights that the impact of CoT is not uniform and can vary significantly even across different open-source architectures.
>
> These findings align with and extend concurrent research [1] on CoT in multimodal models. That work suggests that **longer reasoning chains can cause models to rely more heavily on their powerful language priors while reducing attention to perceptual inputs**, which helps explain the behaviors we observe.
>
> In conclusion, your suggestion has opened up a fascinating and complex area for discussion. Our initial findings suggest CoT can introduce a difficult trade-off between reasoning and perception. We will add a new section with the full results and this important discussion to our final paper. Thank you again for this excellent and thought-provoking feedback.
>
> [1] Liu, Chengzhi, et al. "More Thinking, Less Seeing? Assessing Amplified Hallucination in Multimodal Reasoning Models." arXiv. 2025.

---

> > ### Comment · Reviewer_w2gh · 2025-08-06
> >
> > Thank you for your response. The additional explanations have addressed my concerns satisfactorily. I will maintain my original rating.

---

> > > ### Author Response · Authors · 2025-08-06
> > >
> > > We sincerely thank you for your positive review and for the thought-provoking suggestion to explore Chain-of-Thought prompting. This led to a new set of experiments and a more nuanced analysis that we believe has greatly strengthened the paper. We welcome any further discussion or questions you may have.

---

### Official Review · Reviewer_resW · 2025-06-15

**Rating:** 5
**Confidence:** 4

**Summary:**

This paper introduces CMM benchmark to systematically evaluate hallucination problems in Large Multimodal Models when processing language, vision, and audio simultaneously. The benchmark utilizes 1,200 multimodal samples, along with 2,400 binary yes/no questions, to quantitatively measure models' tendency to generate incorrect outputs regarding non-existent *objects or events*. The paper identifies two main causes of hallucination: unimodal over-reliance and spurious inter-modality correlations.

**Dataset Code Accessibility:**

Yes

**Ethical Considerations:**

No, there are no or only very minor ethics concerns

**Final Justification:**

I did not have major concern with this paper in my initial review, therefore my score remains positive.

**Limitations Weaknesses:**

While not essential to the paper's core content, conducting simple ablation studies based on the identified causes to verify the diagnosed reasons and identifying potential solutions would have strengthened the paper significantly.

**Strengths Contributions:**

- Presents the first comprehensive 3-modality (language/vision/audio) hallucination benchmark
- Experimentally decomposes and validates key causes such as unimodal over-reliance and spurious inter-modality correlations through specific experiments
- Effectively diagnoses causes by separating them into two axes: Perception Accuracy and Hallucination Resistance
- Proposes a diagnostic framework that identifies the causes of hallucination through noise injection experiments
- The analysis of why different models have varying PA and HR scores through the diagnostic framework is particularly interesting

---

> ### Author Rebuttal · Authors · 2025-07-30
>
> We are very grateful to the reviewer for your positive feedback and this excellent suggestion. We agree that conducting verification studies for our diagnosed causes significantly strengthens our claims. We have followed this advice by highlighting existing analyses in our paper and conducting new analyses which we will add to the final version.
>
> ## **Ablation Studies on Diagnosed Reason Verification**
>
> ### **1. Verification for Unimodal Over-reliance:**
>
> **Ablation via Modality Degradation:** Our analysis on unimodal dominance (qualitative and quantitative analysis in Fig. 2 and Appendix A.1) functions as a direct ablation. We showed that by systematically degrading the dominant modality (e.g., blurring the video), the model's hallucinations stemming from that modality consistently decreased. This verifies that the over-reliance was the direct cause.
> **Targeted Mitigation via Prompting:** We conduct experiments prompting LMMs to focus on a single modality. Specifically, in scenarios where LMMs tend to rely on visual while ignoring audio inputs, we prompt them to focus more on the audio content (e.g., `“Please focus more on the given audio information to answer the question.”` for `“Did you hear…”` queries). A similar pattern also applies to cases where LLMs tend to overrely on audio inputs.
>
> | Model | Visual Dom (pa) | Visual Dom (hr) | Audio Dom (pa) | Audio Dom (hr) |
> | :--- | :--- | :--- | :--- | :--- |
> | **Gemini-1.5-flash** | 79.0 | 36.5 | 90.5 | 86.5 |
> | *+focus prompt* | 94.5 | 29.0 | 95.0 | 68.5 |
> | **GroundingGPT** | 99.5 | 1.0 | 98.5 | 23.5 |
> | *+focus prompt* | 96.0 | 1.5 | 96.5 | 41.5 |
> | **FAVOR** | 89.0 | 21.5 | 92.0 | 43.5 |
> | *+focus prompt* | 88.0 | 30.0 | 89.5 | 57.5 |
> | **VideoLLaMA2** | 62.0 | 75.5 | 92.0 | 43.5 |
> | *+focus prompt* | 67.0 | 52.5 | 83.0 | 81.5 |
>
> This experiment, which showed that prompting can **reduce hallucinations to some extent**, further validates our diagnosis by showing how explicitly **counteracting the diagnosed cause** can influence the model's output. However, we also recognize that this approach is **not realistic for real-world deployment**, as it presupposes prior knowledge of which modality is dominant for a given input, which is impractical in practice.
>
> ### **2. Verification for Spurious Inter-modality Correlations:**
>
> Inspired by your suggestion, we have performed additional analysis to verify this diagnosis. Our hypothesis is that these hallucinations are driven by **surface-level statistical correlations between specific *phrases* in the training data**.
>
> To verify this, we will provide qualitative demonstrations for two spurious correlation sub-categories (Visual-Language and Audio-Language). The principle is as follows:
> * For a case where a model hallucinates a common object, we test if the hallucination persists when the object is referred to by a more uncommon, but semantically identical, synonym.
> * For instance, in a Visual-Language case where a "person" is visible, a model might hallucinate a "dog". When we change the query word from "dog" to "canine," we observe that the hallucination tendency decreases significantly.
> * For an Audio-Language case with audio of a dog whimpering, a model is more likely to hallucinate the common phrase "dog barking" than an uncommon but semantically identical one like "emitting a sharp vocalization."
>
> | Case | Input | Query Type | Query Text | Model's Predicted Probability |
> | :--- | :--- | :--- | :--- | :--- |
> | **Visual-Language** (PandaGPT) | Video of a person | Common Phrase | "Did you see a **dog** in the video?" | **P(Yes): 96.05%**, P(No): 3.95% |
> | | | Uncommon Synonym | "Did you see a **canine** in the video?" | P(Yes): 13.24%, **P(No): 86.76%** |
> | **Audio-Language** (Qwen2-Audio) | Audio of a car door closing | Common Phrase | "Did you hear car **honk** in the audio?" | **P(Yes): 92.89%**, P(No): 7.11% |
> | | | Uncommon Synonym | "Did you hear car **horn's blare** in the audio?"| P(Yes): 36.82%, **P(No): 63.18%** |
>
> The probabilities in the final column, calculated from model's output logits of "yes" and "no" via the softmax function, demonstrate the core finding: a model's high confidence in hallucinating "Yes" for a common, correlated phrase can flip to a high confidence in "No" when the query is changed to a semantically similar but statistically uncommon phrase.
> This result supports our diagnosis that the **hallucination is triggered by superficial word co-occurrence statistics**. Due to the time and space constraints of this rebuttal, we have outlined the methodology here. We have observed this phenomenon across diffrent models and spurious correlation types and will deliver these qualitative examples along with a more thorough quantitative analysis in our final camera-ready version.
>
> ***
>
> ## **Identifying Potential Solutions**
>
> Furthermore, as the reviewer insightfully suggests, our diagnostic findings point toward several potential solutions. While a full implementation is beyond our paper's scope, we outline these promising future directions, which we will add to our manuscript:
>
> ### **1. To Address Unimodal Over-reliance:**
>
> * **Mitigation via Input Manipulation:** Based on our modality degradation analysis, one promising direction involves manipulating inputs for mitigation. Related work has explored this from a fascinating and **complementary angle**. Techniques like Visual Contrastive Decoding (VCD) [1] have found that adding noise to the *non-dominant* modality (e.g., the visual input, when strong language priors are dominant) can actually **amplify hallucinations**. It then **contrasts the output distribution** from the original input against the one from the noised input to effectively suppress the hallucinatory content. This principle has been shown in subsequent works [2, 3] to be a reliable mitigation strategy.
> * **Dynamic Attention Routing:** Our "focus prompting" verification suggests that guiding the model's attention is beneficial. A potential solution, therefore, is to design a **dynamic attention routing mechanism**. Such a mechanism could learn to allocate attention to different modalities depending on the context of the input query, making the fusion process more intelligent and less prone to static biases.
>
> ### **2. To Address Spurious Inter-modality Correlations:**
>
> * **Inference-Time Ensemble via Query Rephrasing:** Building on our verification findings, another potential solution is an inference-time strategy. For a given textual query, one could automatically generate semantically identical alternatives using less common phrases. By combining the output logits from both the original and the rephrased queries, this ensemble approach could mitigate hallucinations, as a high-confidence "No" from an uncommon phrase could **temper the biased**, high-confidence "Yes" from the original query.
> * **Semantic Data Augmentation:** To counter hallucinations from superficial word statistics, one could fine-tune models on datasets with augmented captions. Using rare synonyms and paraphrasing would break surface-level correlations and force the model to **learn deeper, more robust conceptual links**.
> * **Data Filtering and Balancing:** A more foundational approach would be to address the biases in the training data itself. By carefully **filtering and rebalancing** datasets to reduce the frequency of common, spurious co-occurrences, models would be less likely to learn these flawed associations in the first place.
> * **Post-training Alignment:** Finally, established techniques like Reinforcement Learning from Human Feedback (RLHF) can be specifically targeted to solve this issue[4][5][6]. By using human feedback to **penalize** the model for generating factually inconsistent or correlated-but-absent objects, the model can be directly aligned to be more faithful to the multimodal input.
>
>
> We sincerely thank the reviewer for the thoughtful and constructive feedback. Your suggestions have been invaluable, prompting us to not only better articulate the verification studies already present in our work but also to outline a clear roadmap for future research on mitigation. We hope these additions will better highlight CMM's value as both a diagnostic tool and a foundation for future research. We will incorporate these valuable discussions into our final manuscript. Thank you again for your supportive review.
>
>
> [1] Leng, Sicong, et al. "Mitigating object hallucinations in large vision-language models through visual contrastive decoding." CVPR. 2024.
> [2] Chen, Zhaorun, et al. "HALC: object hallucination reduction via adaptive focal-contrast decoding." ICML. 2024.
> [3] An, Wenbin, et al. "Mitigating object hallucinations in large vision-language models with assembly of global and local attention." CVPR. 2025.
> [4] Sarkar, Pritam, et al. "Mitigating Object Hallucination in MLLMs via Data-augmented Phrase-level Alignment." ICLR. 2024.
> [5] Sun, Zhiqing, et al. "Aligning Large Multimodal Models with Factually Augmented RLHF." ACL. 2024.
> [6] Yang, Zhihe, et al. "Mitigating hallucinations in large vision-language models via dpo: On-policy data hold the key." CVPR. 2025.

---

> ### Comment · Reviewer_resW · 2025-08-02
>
> Thank you for your response. My score remains positive.

---

> > ### Author Response · Authors · 2025-08-04
> >
> > Thank you for your insightful review and for your excellent suggestions, which have helped us to significantly improve the paper by adding further verification studies and a discussion on potential solutions. We are grateful for your time in reviewing our response and for your continued support. We would be happy to discuss any further points you may have.

---

### Official Review · Reviewer_mciK · 2025-06-27

**Rating:** 5
**Confidence:** 3

**Summary:**

The authors presented a novel benchmark for evaluating hallucinations of large multimodal models across video, audio and language modalities, called CMM. The authors carefully categorized 6 types of hallucinations (3 of which is over-reliance on a certain modality, and 3 of which are spurious correlation across a certain pair of modalies), and have 400 instances for each situation. The authors experimented several existing proprietary and open-source LMMs on the benchmark, evaluating both perception accuracy and hallucination resistance, and the results indicate that existing models still suffer from hallucination vulnerability across some or all types of hallucinations in the benchmark. Additional analysis also shows that larger models tends to be more resistant to hallucinations compared to smaller models.

**Additional Feedback:**

Typos: In figure 1c, "lightening" should be "lightning"

**Dataset Code Accessibility:**

Yes

**Dataset Code Comments:**

The data is fully available on huggingface. The evaluation scripts are available on Github with link from huggingface.

**Ethical Considerations:**

No, there are no or only very minor ethics concerns

**Final Justification:**

I did not have major concern with this paper in my initial review, therefore my score remains positive.

**Limitations Weaknesses:**

Presentation quality: The font size in the figures (especially Fig 1,2,3) are too small. Consider making them larger.

**Strengths Contributions:**

1. The paper formally defined and illustrated 6 types of hallucinations present in video-based LMMs, and performed quantitative validation experiments on each type.

2. The authors collected a balanced set of videos for all 6 types of hallucinations, with extensive human annotation and verification.

3. The benchmark was experimented with many proprietary and open-source LMMs, and results indicate that the benchmark can indeed reveal SOTA model weaknesses in hallucinations, thus showing the need for future research in reducing LMM hallucinations, and that this benchmark can serve as an indicative evaluation metric during the research progress.

---

> ### Author Rebuttal · Authors · 2025-07-30
>
> We are sincerely grateful for your thoughtful and supportive review. We especially appreciate your insightful summary and recognition of our key contributions. We were particularly encouraged by your comments on our efforts to formally define hallucination types, create a balanced benchmark, and demonstrate the benchmark's effectiveness in revealing the weaknesses of SOTA models, as these were our primary goals.
>
> We also thank the reviewer for the valuable suggestions on improving the paper's presentation. We will address all the points raised in the final camera-ready version:
>
> * The font size in Figures 1, 2, and 3 will be increased for better readability.
> * The typo "lightening" in Figure 1c will be corrected to "lightning".
>
> We also wanted to share that, inspired by the valuable feedback from the review process, we are working to further strengthen the final paper. This includes new evaluations on the latest models (such as the GPT and Qwen series), additional ablation studies to verify our diagnosed causes, and an expanded discussion on potential mitigation strategies. We hope these additions will make our work a more valuable resource for the community, and as a living benchmark, we are committed to maintaining CMM as a living resource and will continue to evaluate new models to provide valuable and timely insights for the research community.
>
> Your support and positive feedback are incredibly encouraging to us. Thank you once again for your time and for this thoughtful review.

---

> > ### Comment · Reviewer_mciK · 2025-08-01
> >
> > Thank you for your response. My score remains positive.

---

> > > ### Author Response · Authors · 2025-08-04
> > >
> > > We are very grateful for your supportive and thoughtful review. We appreciate your positive assessment and your helpful suggestions on improving the paper's presentation, which we will incorporate into the final version. We welcome any further discussion you may wish to have.

---

### Official Review · Reviewer_ViGq · 2025-07-02

**Rating:** 4
**Confidence:** 3

**Summary:**

This paper introduces a new benchmark, CMM (The Curse of Multi-Modalities), to systematically evaluate hallucinations in large multimodal models (LMMs) across language, vision, and audio. The authors identify two key sources of hallucinations, over-reliance on unimodal priors and spurious inter-modal correlations, and demonstrate that current LMMs often hallucinate by favoring one modality over others. CMM offers a structured framework for analyzing such vulnerabilities, with the goal of improving cross-modal learning and reducing hallucinations in real-world scenarios.

**Dataset Code Accessibility:**

Yes

**Ethical Considerations:**

No, there are no or only very minor ethics concerns

**Final Justification:**

I have no further questions appreciate all the responses.

**Limitations Weaknesses:**

While the benchmark addresses a timely and important issue, I do have some concerns regarding its current scope and evaluation choices. First, although the dataset is described as balanced and systematically constructed, comprising 1,200 samples and 2,400 probing questions across language, vision, and audio, the overall scale still feels limited given the complexity and diversity of the three modalities involved.  This becomes particularly noticeable when proprietary models continue to perform well on the metrics, suggesting that further scaling could increase the benchmark’s discriminatory power.

Additionally, I found it surprising that GPT-based models were not included in the evaluation, especially considering their dominance and widespread use in multimodal settings. Instead, the closed-source baselines focus mainly on Gemini, which, while competitive, does not represent the full spectrum of current leading models. Including more widely used models, such as the GPT series or the increasingly popular Qwen family, would improve generalizability and relevance.

Lastly, the paper discusses hallucination mitigation through the introduction of noise into the input, which is an interesting direction, but I’m not entirely convinced about the reliability or objectivity of this method without a clearer formalization. It's unclear how the added noise impacts the preservation of original content, and whether this introduces confounding factors that might skew hallucination evaluation.

**Strengths Contributions:**

The issue of hallucinations in LMMs is both under-explored and increasingly critical as multimodal models gain popularity.  And by highlighting model failures that reflect biases seen in deployed systems (e.g., over-trusting language priors), the benchmark aligns well with practical concerns in LMM deployment.

---

> ### Author Rebuttal · Authors · 2025-07-30
>
> # Response for Q1:
> We sincerely thank the reviewer for your insightful feedback regarding the benchmark's scope and discriminatory power. We appreciate the opportunity to clarify our contribution.
>
> **Regarding the scale of the benchmark:** We focused on diagnostic precision and quality, employing an extensive data construction and manual verification process to ensure the **diversity and accuracy** of our test cases. This meticulous process resulted in a diverse set of challenging cases, with full object/event statistics detailed in Appendix C.
> While prioritizing this careful curation, we also ensured CMM's scale is substantial. With 1,200 videos/video-audio pairs and 2,400 questions, CMM is **comparable to or larger than** existing well-recognized hallucination benchmarks:
> * POPE[1]: 500 images, 3,000 question pairs
> * Event-Hallusion[2]: 400 video-question pairs
> * HallusionBench[3]: 346 images, 1,129 questions
>
> **Regarding the benchmark's discriminatory power:** The core strength of CMM is that it is highly effective at **identifying specific, severe weaknesses** even in state-of-the-art models. Our results validate this **diagnostic power**. While leading proprietary models exhibit high Perception Accuracy (PA), their performance falters on targeted tests of Hallucination Resistance (HR).
> For example, Gemini-1.5-Pro achieves a **strikingly low HR of 14.5** on our Audio-Language (AL) correlation task, and Gemini-1.5-Flash scores only **36.5** in HR for Visual Dominance. Uncovering such **critical failure modes**, which pose serious risks for real-world deployment, is precisely the value CMM provides and confirms its challenging nature.
>
> We will ensure this important context is clarified in the final version. Thank you again for your valuable comments.
>
> [1] Li, Yifan, et al. "Evaluating Object Hallucination in Large Vision-Language Models." EMNLP. 2023.
> [2] Zhang, Jiacheng, et al. "Eventhallusion: Diagnosing event hallucinations in video llms." CoRR. 2024.
> [3] Guan, Tianrui, et al. "HallusionBench: an advanced diagnostic suite for entangled language hallucination and visual illusion in large vision-language models." CVPR. 2024.
>
> ***
>
> # Response for Q2:
> We sincerely thank the reviewer for this excellent suggestion. We agree that including GPT-series and Qwen-series models provides a more comprehensive evaluation and strengthens the paper's conclusions. Following your advice, we have conducted additional experiments.
>
> ## Evaluation of GPT-series Models
> As the GPT models do not support audio inputs, we evaluated three flagship models (GPT-4.1, GPT-4.1 mini, and GPT-4.1 nano [1]) on the relevant visual-language (VL) subsets of our CMM benchmark.
>
> Table 1: Evaluation on the Visual-Language (VL) Correlations Subset
>
> | Model | Granularity | Perception Accuracy (PA) | Hallucination Resistance (HR) |
> | :--- | :--- | :--- | :--- |
> | **GPT-4.1** | Object-level | 0.9697 | 0.9388 |
> | | Event-level | 0.8500 | 0.9600 |
> | **GPT-4.1 mini**| Object-level | 0.9381 | 0.9286 |
> | | Event-level | 0.8400 | 0.9600 |
> | **GPT-4.1 nano**| Object-level | 0.9796 | 0.8283 |
> | | Event-level | 0.8990 | 0.9000 |
>
> Table 2: Evaluation on the Language Dominance Subset
>
> | Model | Granularity | Perception Accuracy (PA) | Hallucination Resistance (HR) |
> | :--- | :--- | :--- | :--- |
> | **GPT-4.1** | Object-level | 0.9000 | 0.9400 |
> | | Event-level | 0.7900 | **0.8100** |
> | **GPT-4.1 mini**| Object-level | 0.8500 | 0.9100 |
> | | Event-level | **0.6800** | **0.7600** |
> | **GPT-4.1 nano**| Object-level | 0.8100 | **0.6300** |
> | | Event-level | **0.7526** | **0.3535** |
>
> While the models perform strongly on the well-studied VL Correlations task, our benchmark was still able to **identify critical, specific vulnerabilities** in the Language Dominance category. Here, we observe two key trends:
> 	1. Event-level queries are more challenging: Across all models, the Hallucination Resistance (HR) is consistently lower for event-level questions compared to object-level ones, highlighting the **difficulty in overcoming temporal, event-based language priors**.
> 	2. Smaller models are more vulnerable: This issue of language dominance becomes **significantly more prevalent** as model size decreases. The HR score for event-level queries drops sharply from 0.8100 (GPT-4.1) to a very **low 0.3535** (GPT-4.1 nano).
>
> This demonstrates the effectiveness of our framework in **pinpointing specific, critical weaknesses** that might be obscured by aggregate metrics, even in highly-capable models.
>
> ## Evaluation of Qwen-series Models
> We have also completed our evaluation of the increasingly popular Qwen family. We tested Qwen2.5-Omni-7B[2], a model released around our submission date that supports full tri-modal inputs, and Qwen2.5-VL-7B[3], which takes visual inputs. The results are presented below.
>
> **Table 3: Evaluation of Qwen-Omni (Full Benchmark)**
>
> | Sub-category | Perception Accuracy (PA) | Hallucination Resistance (HR) |
> | :--- | :--- | :--- |
> | VL Correlations | 0.8850  | 0.9700  |
> | AL Correlations | 0.9200  | 0.8350  |
> | VAL Correlations | 0.9100  | 0.9750  |
> | Visual Dominance | 0.8950  | **0.7450**  |
> | Audio Dominance | 0.8250  | 0.8000  |
> | Language Dominance | **0.6850**  | 0.8500  |
>
> **Table 4: Evaluation of Qwen-VL 2.5 (VL Subsets)**
>
> | Sub-category | Perception Accuracy (PA) | Hallucination Resistance (HR) |
> | :--- | :--- | :--- |
> | VL Correlations | 0.8900  | 0.9700  |
> | Language Dominance | **0.6650**  | 0.8700  |
>
> The Qwen results further validate our benchmark's utility. The powerful Qwen-Omni model, while achieving high Hallucination Resistance (HR) on correlation tasks, shows **relative weakness in dominance categories**, particularly Visual Dominance (HR: 0.7450). Similarly, Qwen-VL 2.5 shows a noticeable performance gap between its strong resistance to VL correlations (HR: 0.9700) and its weaker resistance to Language Dominance (HR: 0.8700).
>
> A particularly interesting finding is in the Language Dominance category. While both models seem to have been optimized for high Hallucination Resistance, this may have come at a **steep cost to their factual grounding**. Both Qwen-Omni and Qwen-VL 2.5 exhibit low Perception Accuracy (PA) scores of 0.6850 and 0.6650, respectively, in this category. This demonstrates the effectiveness of our benchmark in **identifying specific, crucial vulnerabilities and trade-offs** that might otherwise be missed. This reinforces our finding that modality over-reliance remains a complex challenge even for the latest models.
>
> Thank you again for pushing us to broaden our evaluation. As a benchmark paper, we are **committed to maintaining CMM as a living resource** and will **continue to evaluate new models** to provide valuable and timely insights for the research community.
>
> [1] OpenAI. ChatGPT, GPT‑4.1 version. https://openai.com/index/gpt-4-1/. 2025.
> [2] Xu, Jin, et al. "Qwen2.5-Omni Technical Report". ArXiv. 2025.
> [3] Bai, Shuai, et al. "Qwen2.5-VL Technical Report". ArXiv. 2025.
>
> ***
>
> # Response for Q3:
> We thank the reviewer for your insightful question and for highlighting this interesting direction. We agree that the reliability and objectivity of any proposed mitigation method are paramount, and we appreciate the chance to clarify the purpose of our noise-based analysis.
>
> The experiments in our paper where we introduce noise or blur (e.g., Fig. 2 , Appendix A.1 ) were intended as a **diagnostic case study** to **validate our hypothesis of unimodal over-reliance, rather than as a formal mitigation proposal**. Our analysis showed that by degrading information from the dominant modality (e.g., blurring the video in a visual dominance scenario), the model's tendency to hallucinate based on that modality decreases. This directly supports our claim about the cause of these specific failures.
>
> That being said, we strongly agree that this is an interesting direction for hallucination mitigation, and related work has explored this from a fascinating and **complementary angle**. Techniques like Visual Contrastive Decoding (VCD)[1] have found that adding noise to the non-dominant modality (e.g., the visual input, when strong language priors are dominant) can actually **amplify hallucinations**. It then **contrasts the output distribution** from the original input against the one from the noised input to effectively suppress the hallucinatory content while preserving factual information from the original input. This and subsequent works[2][3] have shown that carefully manipulating input modalities can be a very effective and reliable mitigation strategy.
>
> We also wish to respectfully point the reviewer to our response to `Reviewer resW`. Inspired by similar constructive feedback, we have provided more extensive ablation studies to verify our diagnosed causes and have outlined several promising future research directions for mitigation (some of these directions build upon the ideas we initially presented in Appendix D of our submission).
>
> We hope this clarifies the **diagnostic role** of the experiments in our paper. Your insightful question has helped us not only to better articulate our contributions but also to strengthen the paper by connecting our diagnostic findings to the broader landscape of mitigation research. We are sincerely grateful for your valuable feedback and will ensure these clarifications are made in the final manuscript. Thank you again.
>
> [1] Leng, Sicong, et al. "Mitigating object hallucinations in large vision-language models through visual contrastive decoding." CVPR. 2024.
> [2] Chen, Zhaorun, et al. "HALC: object hallucination reduction via adaptive focal-contrast decoding." ICML. 2024.
> [3] An, Wenbin, et al. "Mitigating object hallucinations in large vision-language models with assembly of global and local attention." CVPR. 2025.

---

> > ### Comment · Reviewer_ViGq · 2025-08-06
> >
> > Thanks for the detailed response. I have no further questions and tend to maintain my score.

---

> > > ### Author Response · Authors · 2025-08-06
> > >
> > > Thank you for taking the time to review our response and for confirming that your initial concerns have been addressed. We are sincerely grateful for your initial feedback, as it directly led to substantial improvements in the paper, including new model evaluations and clearer analyses.
> > >
> > > With this in mind, we would be very appreciative if you would consider whether these improvements might merit a re-evaluation of the score. We fully respect your final decision and thank you again for your invaluable and constructive engagement throughout this process.

---

### Decision · Program_Chairs · 2025-09-18

**Decision:**

Accept (poster)

**Comment:**

This paper introduces The Curse of Multi-Modalities (CMM), the first benchmark to systematically evaluate hallucinations in Large Multimodal Models (LMMs) across language, vision, and audio. By identifying key failure modes - unimodal over-reliance and spurious inter-modality correlations - the authors provide a timely and significant contribution to a critical, under-explored area of LMM reliability.

Reviewers were unanimously positive, commending the novelty and quality of the benchmark and its insightful diagnostic approach. They found the analysis and experiments convincing, agreeing that the work successfully highlights critical vulnerabilities even in state-of-the-art models.

The authors’ proactive and diligent rebuttal further strengthened the submission. They addressed all major reviewer feedbacks by expanding the evaluation to include GPT and Qwen models and conducting new analyses on Chain-of-Thought prompting, enriching the paper's findings and demonstrating a commitment to thoroughness.

In summary, this is a technically sound paper presenting a high-quality, relevant benchmark. The dataset is well-constructed, the findings are significant, and the authors' engagement during the review process was excellent. This paper promises to become an impactful resource for the community, and I am happy to recommend its acceptance.

===== FINAL UPDATE FROM DB Track PCs ====

The final decision for this paper has been taken by the program chairs after consultation with the SACs. All Senior Area Chairs have ranked papers according to the feedback from the AC during the review process. We decided to leave the original meta-review to reflect the opinion of the AC in light of the initial discussions with reviewers and SAC.